# SHAP-XRT: The Shapley Value Meets Conditional Independence Testing

**Jacopo Teneggi**\*                                                              *jtenegg1@jhu.edu*
*Department of Computer Science, Johns Hopkins University*
*Mathematical Institute for Data Science (MINDS), Johns Hopkins University*

**Beepul Bharti**\*                                                              *bbharti1@jhu.edu*
*Department of Biomedical Engineering, Johns Hopkins University*
*Mathematical Institute for Data Science (MINDS), Johns Hopkins University*

**Yaniv Romano**                                                              *yromano@technion.ac.il*
*Departments of Electrical Engineering and of Computer Science, Technion—Israel Institute of Technology*

**Jeremias Sulam**                                                              *jsulam1@jhu.edu*
*Department of Biomedical Engineering, Johns Hopkins University*
*Mathematical Institute for Data Science (MINDS), Johns Hopkins University*

(\*) **Equal contribution**

**Reviewed on OpenReview:** **https://openreview.net/forum?id=WFtTpQ47A7**

## Abstract

The complex nature of artificial neural networks raises concerns on their reliability, trustworthiness, and fairness in real-world scenarios. The Shapley value—a solution concept from game theory—is one of the most popular explanation methods for machine learning models. More traditionally, from a statistical perspective, feature importance is defined in terms of conditional independence. So far, these two approaches to interpretability and feature importance have been considered separate and distinct. In this work, we show that Shapley-based explanation methods and conditional independence testing are closely related. We introduce the **SHAP**ley E**X**planation **R**andomization **T**est (SHAP-XRT), a testing procedure inspired by the Conditional Randomization Test (CRT) for a specific notion of *local* (i.e., on a sample) conditional independence. With it, we prove that for binary classification problems, the marginal contributions in the Shapley value provide lower and upper bounds to the expected *p*-values of their respective tests. Furthermore, we show that the Shapley value itself provides an upper bound to the *p*-value of a *global* (i.e., overall) null hypothesis. As a result, we further our understanding of Shapley-based explanation methods from a novel perspective and characterize the conditions under which one can make statistically valid claims about feature importance via the Shapley value.

## 1 Introduction

Deep learning models have shown remarkable success in solving a wide range of problems, from computer vision and natural language processing, to reinforcement learning and scientific research (LeCun et al., 2015). These exciting results have come hand-in-hand with an increase in the complexity of these models, mostly based on neural networks. While these systems consistently set the state-of-the-art in many tasks, our understanding of their specific mechanisms remains intuitive at best. In fact, as neural networks keep getting deeper and wider, they also become *opaque* (or unintelligible) to both developers and users. This lack of transparency is usually referred to as the *black-box* problem: the predictions of a deep learning model are not readily interpretable, entailing theoretical, societal, and regulatory issues (Zednik, 2021; Tomsett et al.,

2018; Shin, 2021). For example, certain regulations may require companies and organizations who rely on autonomous systems to provide explanations of their decisions (e.g., what lead a model to reject a loan application) (Casey et al., 2019; Kaminski & Malgieri, 2020; Hacker et al., 2020). Finally, it is still unclear whether a loss of interpretability is unavoidable to increase performance (Rudin, 2019).

These concerns on the reliability and trustworthiness of machine learning have motivated recent efforts on explaining predictions. A distinction exists between *inherently* interpretable models and *explanation* methods. The former are designed to provide predictions that can be understood by their users. For example, rule-based systems and decision trees are considered inherently interpretable because it is possible to trace a prediction through the specific rules or splits learned by the model. On the other hand, when models are not explicitly designed in an interpretable fashion (e.g., deep neural networks), we usually rely on *a-posteriori* explanation methods to find the most important features towards a prediction. These methods assign an importance score to every feature (or groups thereof) in the input, and the resulting scores can be presented to a user as an *explanation* of the prediction. Explanation methods have been widely adopted given their easy implementation and their ability to explain any model, i.e. they can be *model-agnostic*. Since the introduction of CAM (Zhou et al., 2016), DeepLift (Shrikumar et al., 2017), and LIME (Ribeiro et al., 2016), the explainability literature has witnessed remarkable growth, and various approaches have been proposed to identify important features (Selvaraju et al., 2017; Lundberg & Lee, 2017; Chen et al., 2018; Covert et al., 2021; Kolek et al., 2021; Jiang et al., 2021; Kolek et al., 2022; Teneggi et al., 2022a; Mosca et al., 2022) with varying degrees of success (Adebayo et al., 2018; Shah et al., 2021).

The SHAP framework (Lundberg & Lee, 2017) unifies several existing methods while providing explanations that satisfy some desirable theoretical properties. More precisely, it brings the Shapley value (Shapley, 1951)—a *solution concept* from game theory (Owen, 2013)—to bear in the interpretation of machine learning predictors. In cooperative game theory, a solution concept is a formal rule that describes the strategy that each player will use when participating in a game. For Transfer Utility (TU) games, the Shapley value is the only solution concept that satisfies the properties of additivity, nullity, symmetry, and linearity, and it can be derived axiomatically.[1] In the context of explainability, the SHAP framework considers a TU game represented by the predictive model, where every feature in the input is a *player* in the game, and it computes their Shapley values as a measure of feature importance.[2] We remark that the axiomatic characterization of the Shapley value implies that it is the only explanation method that satisfies these theoretical properties. The most important limitation of the Shapley value is its exponential computational cost in the number of players (i.e., the input dimension), which quickly becomes intractable for several applications of interest. Thus, most Shapley-based methods rely on various strategies to approximate them (Chen et al., 2023), some with finite-sample, non-asymptotic, and efficiency guarantees (Chen et al., 2018; Lundberg et al., 2020; Teneggi et al., 2022a). An alternative approach is to relax some of the axioms. This yields several other explanation methods based on a feature's marginal contribution, such as semivalues (Dubey et al., 1981) and random order values (Frye et al., 2020b).

While theoretically appealing, and often useful in practice, this Shapley-based approach to interpretability remains questionable:

*What does it mean for a feature to receive a large Shapley value?*

Previous works have explored the connection between feature selection and the Shapley value (Cohen et al., 2007; Fryer et al., 2021; Rozemberczki et al., 2022) as well as some statistical implications of large (or small) Shapley values (Owen & Prieur, 2017; Hooker, 2007; Ma & Tourani, 2020; Verdinelli & Wasserman, 2021; 2023). However, a general and precise connection between the Shapley value and statistical importance has been missing. In particular, it is unclear how to provide false positive rate control (i.e., to control the Type I error) when feature selection is performed on the basis of Shapley values. This is a fundamental issue as explanation methods are increasingly used in high-stakes scenarios to inform decision making, especially in medicine (Zoabi et al., 2021; de Almeida et al., 2022; Liu & Xie, 2020). For example, consider a clinical setting where a predictor is used to support diagnosis and treatment. Given each feature's Shapley value,

---

[1]In a TU game, players can exchange their utility without incurring in any cost.
[2]Herein, we will implicitly refer to the *Shapley value* of a feature according to the definition of SHAP attribution in Lundberg & Lee (2017).

which ones should we deem important? If a patient's age, blood pressure, weight, and height receive Shapley values of 0.30, 0.15, 0.35, and 0.20, respectively, which should we report? We could decide to report the most important feature (i.e., weight). This choice will likely underreport findings and potentially mislead the clinician. Symmetrically, reporting all features with a positive Shapley value would be uninformative with respect to the decision process of the model. This example shows the lack of a rigorous way to perform statistically-valid feature selection by means of the Shapley value, and how overlooking this question could lead to potentially harmful consequences in real-world scenarios.

In this paper, we show how relating the Shapley value to statistical notions of conditional independence provides a principled way to select important features. Indeed, the statistics literature has a rich history of studying variable selection problems (Blum & Langley, 1997; James et al., 2013; Guyon & Elisseeff, 2003), for example, via conditional independence testing and controlled variable selection (Candes et al., 2018). That is, a feature is unimportant if it is independent of a response once conditioned on the remaining features. In other words, knowing the value of an unimportant feature does not provide any more information about the response when the rest of the data is known. The Conditional Randomization Test (CRT) (Candes et al., 2018) is a conditional independence test that does not make any assumptions on the conditional distribution of the response given the features, while assuming that the conditional distribution of the features is known instead. This setting is particularly useful in applications where unlabeled data may be abundant compared to labeled data (e.g., genomics research, as shown in (Candes et al., 2018; Sesia et al., 2021; Sesia & Sun, 2022), or imaging data (Nguyen et al., 2019)). While the original CRT procedure is computationally intractable for large models, fast and efficient alternatives have been recently proposed, such as the the Holdout Randomization Test (HRT) by Tansey et al. (2022) and the Distilled Conditional Randomization Test (DCRT) by Liu et al. (2022).

We remark that in this paper we consider feature importance with respect to the response of a fixed predictive model on an *individual* sample. This differs from the traditional statistical setting in which one analyzes features with respect to an observed response at the population level. Nonetheless, this notion of interpretability is important in many scenarios. For example, one may wish to understand the important features for a model that computes credit scores (Demajo et al., 2020; Bücker et al., 2022), or one may need to verify that the important features for the prediction of an existing, complex model agree with prior-knowledge (Burns et al., 2020; Jiang et al., 2021; Fong & Vedaldi, 2017).

## 1.1 Related work

Previous works have explored granting the Shapley value with some statistical notion of importance. For example, Owen & Prieur (2017) and Verdinelli & Wasserman (2021; 2023) study the Shapley value for correlated variables in the context of ANalysis Of VAriance (ANOVA) (Hooker, 2007) and Leave Out COvariates (LOCO) (Lei et al., 2018) procedures, respectively. Importantly, Verdinelli & Wasserman (2021) precisely raise the issue of the lack of statistical meaning for large (or small) Shapley values when features are correlated. Information-theoretic interpretations of the Shapley value have also been proposed. SAGE (Covert et al., 2020) translates the Shapley value from local (i.e., on a sample) to global (i.e., over a population) feature importance, and shows connections to mutual information. Most recently, (Watson et al., 2023) propose a modified Shapley value with precise information-theoretical properties to study the independence between the true response and the features. In the context of data attribution methods, Ghorbani et al. (2020) define the *Distributional Shapley* value to include information about the underlying data distribution into the original Shapley value. Finally, and most closely to this work, Ma & Tourani (2020) try and deploy ideas of conditional independence to the Shapley value for causal inference on data generated by a Bayesian network. Our work differs from these, as we now summarize.

## 1.2 Contributions

In this paper, we will show that for machine learning models, computing the Shapley value of a given input feature amounts to performing a series of conditional independence tests for specific feature importance definitions. This is different from Watson et al. (2023) in that we do not assume the predictor to be the true conditional distribution of the response given the features. As we will shortly demonstrate, the computed

game-theoretic quantities provide upper and lower bounds to the *p*-values of their respective tests. This novel connection provides several popular explanation techniques with a well-defined notion of statistical importance. We demonstrate our results with simulated as well as real imaging data. We stress that we do not wish to introduce a novel explanation method for model predictions. Our aim is instead to expand our fundamental understanding of the Shapley value when applied to machine learning models, particularly as its use for the interpretation of predictive models continues to grow (Moncada-Torres et al., 2021; Zoabi et al., 2021; Liu et al., 2021). These novel insights should inform the design of explanation methods to guarantee the responsible use of machine learning models.

## 2 Background

Before presenting the contribution of this work, we briefly introduce the necessary notation and general definitions. Herein, we will denote random variables and random vectors with capital letters (e.g., $X$), their realizations with lowercase letters (e.g., $X = x$) and random matrices with bold capital letters (e.g., $\boldsymbol{X}$). Let $\mathcal{X}, \mathcal{Y}$ be some input and response domains, respectively, such that $(X, Y) \sim \mathcal{D}$ is a random sample from an unknown distribution $\mathcal{D}$ over $\mathcal{X} \times \mathcal{Y}$. We set ourselves within the classical binary classification setting. Given $\mathcal{X} \subset \mathbb{R}^n$, $\mathcal{Y} = \{0, 1\}$, the task is to estimate the binary response $Y$ on a sample $X$ by means of a predictor $f : \mathcal{X} \to \mathcal{Y}$ that approximates the conditional expectation of the response on an input, i.e. $f(x) \approx \mathbb{E}[Y \mid X = x]$. We will not concern ourselves with this learning problem. Instead, we assume we are given a fixed predictor which we wish to analyze. More precisely, for an input $x \in \mathbb{R}^n$, we want to understand the importance of the features in $x$, i.e. $x_i$, with respect to the value of $f(x)$. This question will be formalized by means of the Shapley value, which we now define.

### 2.1 The Shapley value

In game theory (see, e.g., (Peleg & Sudhölter, 2007)), the tuple $([n], v)$ represents an $n$-person cooperative game with players $[n] := \{1, \ldots, n\}$ and characteristic function $v : \mathcal{P}(n) \to \mathbb{R}^+$, where $\mathcal{P}(n)$ is the power set of $[n]$. For any subset of players $C \subseteq [n]$, the characteristic function outputs a nonnegative score $v(C)$ that represents the utility accumulated by the players in $C$. Furthermore, one typically assumes $v(\emptyset) = 0$. A Transfer Utility (TU) game is one where players can exchange their utility without incurring any cost. Given a TU game $([n], v)$, a *solution concept* is a formal rule that assigns to every player in the game a reward that is commensurate with their contribution. In particular, there exists a unique solution concept that satisfies the axioms of additivity, nullity, symmetry, and linearity (Shapley, 1951) (see Appendix B for details on the axioms). This solution concept is the set of Shapley values, $\phi_1([n], v), \ldots, \phi_n([n], v)$, which are defined as follows.

**Definition 1** (Shapley value). Given an $n$-person TU game $([n], v)$, the Shapley value of player $j \in [n]$ is

$$\phi_j([n], v) = \sum_{C \subseteq [n] \setminus \{j\}} w_C \cdot [v(C \cup \{j\}) - v(C)], \tag{1}$$

where $w_C = 1/n \cdot \binom{n-1}{|C|}^{-1}$. That is, $\phi_j([n], v)$ is the average marginalized contribution of the $j^{\text{th}}$ player over all subsets (i.e., coalitions) of players.

### 2.2 Explaining model predictions with the Shapley value

Definition 1 shows how to compute the Shapley value for *players* in a TU game, and it is not immediately clear how this would apply to machine learning models. To this end, let $f(x)$ be the prediction of a learned model on a new sample $x \in \mathbb{R}^n$. For any set of features $C \subseteq [n]$, denote $x_C \in \mathbb{R}^{|C|}$ the entries of $x$ in the subset $C$ (analogously, $x_{-C} \in \mathbb{R}^{n-|C|}$ for $-C := [n] \setminus C$, the complement of $C$). We will refer to

$$\widetilde{X}_C = [x_C, X_{-C}] \in \mathbb{R}^n \tag{2}$$

as the random vector that is equal to $x$ in the features in $C$ and that takes a *random reference* value in its complement $-C$. Following Lundberg & Lee (2017), we let $X_{-C}$ be sampled from its conditional distribution

given $x_C$, i.e. $X_{-C} \mid X_C = x_C$.[3] Note that, in this way, $X_{-C}$ is independent of the response $f(x)$ by construction. For the sake of simplicity, we write $\widetilde{X}_C \sim \mathcal{D}_{X_C = x_C}$. Then, for a predictor $f$ and every subset $C \subset [n]$, $f(\widetilde{X}_C)$ is a random variable whose source of randomness is the random reference value.[4] This way, with abuse of notation, one can define the TU game $(x, f)$, where every feature in the sample $x$ is a player in the game. Therefore, analogous to Definition 1, the Shapley value of feature $j \in [n]$ is

$$\phi_j(x, f) = \sum_{C \subseteq [n] \setminus \{j\}} w_C \cdot \mathbb{E}[f(\widetilde{X}_{C \cup \{j\}}) - f(\widetilde{X}_C)]. \tag{3}$$

We note that $\phi_j(x, f)$ can be negative, i.e. feature $j$ has an overall negative contribution to the prediction of the model. This differs from the game-theoretical setting where feature attributions are usually considered nonnegative. In the context of explainability, we look for a subset $C^* \subseteq [n]$ (e.g., the smallest subset) such that $x_{C^*}$ are the observed features that contributed the most towards $f(x)$.

We remark two main differences between the original definition of the Shapley value from game theory (Equation (1)) and its translation to machine learning models (Equation (3)) in common settings: $(i)$ the characteristic function $v$ can predict on sets of players, whereas the predictive model $f$ typically has a fixed input domain $\mathcal{X} \subset \mathbb{R}^n$ (e.g., for a convolutional neural network);[5] and $(ii)$ the characteristic function $v$ is usually not data-dependent, while in this machine learning setting the model $f$ was trained on samples from a specific distribution $\mathcal{D}$. As a result, given a subset of features $C \subseteq [n]$, one cannot simply predict on the partial input vector $x_C \in \mathbb{R}^{|C|}$ because $f(x_C)$ is not defined. Instead, the missing features in the complement of $C$ must be masked with a reference value that does not contain any information about the response in order to simulate their absence. Furthermore, this reference value must come from the same data distribution $\mathcal{D}$ the model was trained on, hence the need to sample from the conditional distribution $X_{-C} \mid X_C = x_C$ (Frye et al., 2020a; Aas et al., 2021). Both the need to approximate the conditional distribution as well as the exponential number of summands in Equation (3) make the exact computation of Shapley values intractable in general.

## 2.3 The Conditional Randomization Test (CRT)

Before introducing our novel conditional independence testing procedure, we provide a brief summary of feature importance from the perspective of conditional independence. Given random variables $X, Y$, and $Z$ we say that $X$ is independent of $Y$ given $Z$ (succinctly, $X \perp\!\!\!\perp Y \mid Z$) if $(X \mid Y, Z) \stackrel{d}{=} (X \mid Z)$, where $\stackrel{d}{=}$ indicates equality in distribution. Put into words, knowing $Y$ does not provide any more information about $X$ in the presence of $Z$. Recall that for each $j \in [n]$, $X_j \in \mathbb{R}$ is a random variable that corresponds to the $j^{\text{th}}$ feature in $X$, and $X_{-j} \in \mathbb{R}^{n-1}$, $-j := [n] \setminus \{j\}$ is its complement. Then, the CRT procedure by Candes et al. (2018) implements a conditional independence test for the null hypothesis

$$H_{0,j}^{\text{CRT}} : \ Y \perp\!\!\!\perp X_j \mid X_{-j}, \tag{4}$$

which can be directly rewritten as[6]

$$H_{0,j}^{\text{CRT}} : \ (Y \mid X_j, X_{-j}) \stackrel{d}{=} (Y \mid X_{-j}). \tag{5}$$

Let $X_j^{(k)} \sim X_j \mid X_{-j}, \ k = 1, \ldots, K$ be $K$ *null* duplicates of $X_j$. $X_j^{(k)}$ are called nulls because, by construction, they are conditionally independent of the response $Y$ given $X_{-j}$. Under $H_{0,j}^{\text{CRT}}$, for any choice of test statistic $T(X_j, Y, X_{-j})$, the random variables $T(X_j, Y, X_{-j}), T(X_j^{(1)}, Y, X_{-j}), \ldots, T(X_j^{(K)}, Y, X_{-j})$ are i.i.d., hence exchangeable (Berrett et al., 2020). Then, $\hat{p}_j^{\text{CRT}}$—the $p$-value returned by the CRT—is valid: under $H_{0,j}^{\text{CRT}}$, $\mathbb{P}[\hat{p}_j^{\text{CRT}} \leq \alpha] \leq \alpha, \ \forall \alpha \in [0, 1]$ (see Candes et al., 2018, Lemma F.1). For completeness, Algorithm C.1 summarizes the CRT procedure.

---

[3]Herein, we do not touch upon the discussion on *observational* vs. *interventional* Shapley values, and we refer the interested reader to (Sundararajan & Najmi, 2020; Chen et al., 2020; Merrick & Taly, 2020; Janzing et al., 2020).

[4]When $C = [n]$, $f(\widetilde{X}_C)$ is simply the prediction $f(x)$, which is not random.

[5]As noted in Jain et al. (2022); Covert et al. (2022); Teneggi et al. (2022b): Deep Sets (Zaheer et al., 2017) and Transformers (Vaswani et al., 2017) architectures do not suffer from this limitation.

[6]Note that Equation (5) is equivalent to $H_{0,j}^{\text{CRT}} : \ (X_j \mid Y, X_{-j}) \stackrel{d}{=} (X_j \mid X_{-j})$.

## 3 Hypothesis testing via the Shapley value

Given the above background, we will now show that the Shapley value for machine learning explainability (Equation (3)) is tightly connected with conditional independence testing. Formally, we present the SHAPley EXplanation Randomization Test (SHAP-XRT): a novel modification of the CRT that—differently from the original motivation in Candes et al. (2018)—evaluates conditional independence of the response of a deterministic model $f$ on an individual sample $X = x$, $X \sim \mathcal{D}_{\mathcal{X}}$. That is, SHAP-XRT, akin to the CRT, provides a permutation-based $p$-value to test for the conditional independence of two distributions without making any assumptions on the conditional distribution of the prediction of the model, while requiring access to the conditional distribution of the data. Differently from the CRT, the distributions considered by SHAP-XRT are built on top of a single observation $x$ instead of over a population, and the response is a deterministic function of the input rather than a random variable. As we will present shortly, there is a fundamental difference between the Shapley value and SHAP-XRT; the former computes the average marginal contribution of a feature by differences of conditional expectations, whereas the latter provides a $p$-value for a test of equality in distribution.

### 3.1 The Shapley Explanation Randomization Test (SHAP-XRT)

Going back to our motivating example of an automated system in a clinical setting, we would like to deploy ideas of conditional independence testing to report the most important features for the model's prediction on a patient. For example, we might be interested in knowing whether the response of the model on a specific patient is independent of age when their blood pressure and weight are collected. This way, we could provide precise statistical guarantees on the generated explanations, such as Type I error control. More precisely, this is equivalent to testing for the independence of the response of a model $f$ on a feature $x_j$, $j \in [n]$ when the features in $x_C$, $C \subseteq [n] \setminus \{j\}$ are observed (e.g., $j = \texttt{age}$ and $C = \{\texttt{blood pressure}, \texttt{weight}\}$). We now formalize this question by means of our novel SHAP-XRT null hypothesis.

**Definition 2** (**SHAP**ley **EX**planation **R**andomization **T**est). Let $\mathcal{X} \subset \mathbb{R}^n$ and $f : \mathcal{X} \to [0,1]$ be a fixed predictor for a binary response $Y \in \{0,1\}$ on an input $X \in \mathcal{X}$ such that $(X,Y) \sim \mathcal{D}$. Given a new sample $X = x$, $X \sim \mathcal{D}_{\mathcal{X}}$, a feature $j \in [n]$, and a subset of features $C \subseteq [n] \setminus \{j\}$, denote $\widetilde{X}_C = [x_C, X_{-C}]$ the corresponding random vector where $X_{-C}$ is sampled from its conditional distribution given $X_C = x_C$. Then, the SHAP-XRT null hypothesis is

$$H_{0,j,C}^{\text{S-XRT}} : \ f(\widetilde{X}_{C \cup \{j\}}) \stackrel{d}{=} f(\widetilde{X}_C). \tag{6}$$

Recall that $\widetilde{X}_{C \cup \{j\}}$ and $\widetilde{X}_C$ are random vectors equal to $x$ in the features in $C \cup \{j\}$ and $C$, respectively, and that take unimportant random reference values in their complements (Equation (2)). Then, the SHAP-XRT null hypothesis is equivalent to

$$\left( f(\widetilde{X}_{C \cup \{j\}}) \mid X_{C \cup \{j\}} = x_{C \cup \{j\}} \right) \stackrel{d}{=} \left( f(\widetilde{X}_C) \mid X_C = x_C \right), \tag{7}$$

which precisely asks whether the distribution of the response of the model changes when feature $j$ is added to the subset $C$.

How should one test for this null hypothesis? Denote $\widetilde{\boldsymbol{X}}_{C \cup \{j\}} = (\widetilde{X}_{C \cup \{j\}}^{(1)}, \ldots, \widetilde{X}_{C \cup \{j\}}^{(L)}) \in \mathbb{R}^{L \times n}$ and $\widetilde{\boldsymbol{X}}_C = (\widetilde{X}_C^{(1)}, \ldots, \widetilde{X}_C^{(L)}) \in \mathbb{R}^{L \times n}$ the random matrices containing $L$ duplicates of $\widetilde{X}_{C \cup \{j\}}$ and $\widetilde{X}_C$, respectively, such that $\widetilde{Y}_{C \cup \{j\}} = f(\widetilde{\boldsymbol{X}}_{C \cup \{j\}})$ and $\widetilde{Y}_C = f(\widetilde{\boldsymbol{X}}_C)$ are the predictions of the model. Algorithm 1 implements the SHAP-XRT testing procedure and it computes the respective $p$-value, $\hat{p}_{j,C}^{\text{S-XRT}}$, for any choice of test statistic $T$ on the response of the model, e.g. the mean. We now formally state the validity of this test.

**Theorem 1** (Validity of $\hat{p}_{j,C}^{\text{S-XRT}}$). *Under the null hypothesis $H_{0,j,C}^{S\text{-}XRT}$, $\hat{p}_{j,C}^{S\text{-}XRT}$—the p-value returned by the SHAP-XRT procedure—is valid for any choice of test statistic $T$, i.e. $\mathbb{P}[\hat{p}_{j,C}^{S\text{-}XRT} \leq \alpha] \leq \alpha$, $\forall \alpha \in [0,1]$.*

We defer the brief proof to Appendix A.1.

---

**Algorithm 1** Shapley Explanation Randomization Test (SHAP-XRT)

---

**procedure** SHAP-XRT(model $f : \mathbb{R}^n \to [0, 1]$, sample $x \in \mathbb{R}^n$, feature $j \in [n]$, subset $C \subseteq [n] \setminus \{j\}$, test statistic $T$, number of null draws $K \in \mathbb{N}$, number of reference samples $L \in \mathbb{N}$)

   Sample $\widetilde{\boldsymbol{X}}_{C \cup \{j\}} \sim \left( \mathcal{D}_{X_{C \cup \{j\}} = x_{C \cup \{j\}}} \right)^L$

   $\widetilde{Y}_{C \cup \{j\}} \leftarrow f(\widetilde{\boldsymbol{X}}_{C \cup \{j\}})$

   Compute the test statistic $t \leftarrow T(\widetilde{Y}_{C \cup \{j\}})$

   **for** $k \leftarrow 1, \dots, K$ **do**

      Sample $\widetilde{\boldsymbol{X}}_C^{(k)} \sim (\mathcal{D}_{X_C = x_C})^L$

      $\widetilde{Y}_C^{(k)} \leftarrow f(\widetilde{\boldsymbol{X}}_C^{(k)})$

      Compute the null statistic $\tilde{t}^{(k)} \leftarrow T(\widetilde{Y}_C^{(k)})$

   **end for**

   **return** A (one-sided) $p$-value

$$\hat{p}_{j,C}^{\text{S-XRT}} = \frac{1}{K+1} \left( 1 + \sum_{k=1}^{K} \mathbf{1}[\tilde{t}^{(k)} \geq t] \right)$$

**end procedure**

---

We remark that the $p$-value returned by the SHAP-XRT procedure—akin to other conditional independence tests—is one-sided. Rejecting $H_{0,j,C}^{\text{S-XRT}}$ implies that in the presence of the features in $C$, the observed value of feature $j$ is important in the sense that $f(\widetilde{X}_{C \cup \{j\}})$ is often *larger* (i.e., closer to 1), or stochastically greater, than $f(\widetilde{X}_C)$. If one desires to reject $H_{0,j,C}^{\text{S-XRT}}$ for features whose $f(\widetilde{X}_{C \cup \{j\}})$ is often *smaller* than $f(\widetilde{X}_C)$, it suffices to use $1 - f$ in lieu of $f$. Note that the SHAP-XRT procedure encompasses multi-class classification settings where $g : \mathbb{R}^n \to \Delta^M$ predicts one of $M$ classes and $f(x) = g(x)_m$, $m \in [M]$. Furthermore, $H_{0,j,C}^{\text{S-XRT}}$ can be generalized to any predictor $f : \mathcal{X} \to \mathcal{Y}$ for arbitrary domains, but we consider binary classification in this work.

We stress that in contrast with the CRT null hypothesis (Equation (4)), $H_{0,j,C}^{\text{S-XRT}}$ is defined *locally* on a specific input $x$ rather than over a population. Differently from the Interpretability Randomization Test (IRT) by Burns et al. (2020) ($i$) $H_{0,j,C}^{\text{S-XRT}}$ asks for the conditional independence of the response with respect to a single feature $j \in [n]$ rather than a collection of subsets of features, and ($ii$) it allows to condition on arbitrary subsets $C \subseteq [n] \setminus \{j\}$ of features instead of restricting to $C = -j$, i.e. all features but $j$.

Finally, let us make a remark about the involved distributions. Recall that the SHAP-XRT procedure is defined locally on a sample $x$. Intuitively, under the null $H_{0,j,C}^{\text{S-XRT}}$, the distribution of the response of the model $f$ is independent of $x_j$ (the *observed* value of the $j^{\text{th}}$ feature in $x$) conditionally on $X_C = x_C$. Since $f$ is a deterministic function of its input, it is natural to ask when the random variable $f(\widetilde{X}_{C \cup \{j\}})$ has a degenerate distribution. Suppose that for each $C \subseteq [n] \setminus \{j\}$, $X_{-C} \mid X_C = x_C$ is not degenerate (e.g., it is not constant), otherwise $f(\widetilde{X}_{C \cup \{j\}})$ is trivially degenerate. For $C = -j$ (i.e., all features but the $j^{\text{th}}$ one), $\widetilde{X}_{C \cup \{j\}} = x$ because $C \cup \{j\} = [n]$ by definition of the complement set $-(C \cup \{j\})$ is empty, i.e. no reference value is sampled. It follows that for this choice of $C$, $f(\widetilde{X}_{C \cup \{j\}})$ is point mass at $f(x)$, and SHAP-XRT retrieves the IRT procedure (Burns et al., 2020).

### 3.2 Connecting the Shapley value and the SHAP-XRT conditional independence test

We now draw a precise connection between the SHAP-XRT testing procedure and the Shapley value for machine learning explainability. This novel relation furthers our fundamental understanding of Shapley-based explanation methods, and, importantly, it clarifies under which conditions one can (and cannot) make statistically valid claims about feature importance. We stress that—as we will discuss—this conditional independence interpretation is not intended to overcome the well-known computational limitations of the Shapley value nor to replace existing methods. Rather, it offers a previously overlooked perspective, and it

suggests new strategies to develop more powerful, statistically well-grounded alternatives. To begin with, let $T$ in the SHAP-XRT procedure be the identity and set $L = 1$ (we will discuss the effect of $L$ on our results later). Then, for a sample $x$, a feature $j \in [n]$, and a subset $C \subseteq [n] \setminus \{j\}$, the SHAP-XRT test and null statistics become $t = f(\widetilde{X}_{C \cup \{j\}})$ and $\tilde{t} = f(\widetilde{X}_C)$, respectively. Denote

$$\Gamma_{j,C} \coloneqq f(\widetilde{X}_{C \cup \{j\}}) - f(\widetilde{X}_C) \tag{8}$$

the random marginal contribution of feature $j$ to the subset $C$ such that

$$p_{j,C}^{\text{S-XRT}} \coloneqq \mathbb{E}[\hat{p}_{j,C}^{\text{S-XRT}}] \qquad \text{and} \qquad \gamma_{j,C}^{\text{SHAP}} \coloneqq \mathbb{E}[\Gamma_{j,C}] \tag{9}$$

are the expected $p$-value of the SHAP-XRT procedure and the expected marginal contribution, respectively. Note the expectations are over both $\widetilde{X}_{C \cup \{j\}}$ and $\widetilde{X}_C$. This is to take into account the randomness in the draw of test statistic $t$. Then, it is easy to see that the Shapley value of feature $j$ can be rewritten as

$$\phi_j(x, f) = \sum_{C \subseteq [n] \setminus \{j\}} w_C \cdot \gamma_{j,C}^{\text{SHAP}}. \tag{10}$$

We now show that each summand $\gamma_{j,C}^{\text{SHAP}}$ can be used to construct both upper and lower bounds to $p_{j,C}^{\text{S-XRT}}$.

**Theorem 2.** *Let $\mathcal{X} \subset \mathbb{R}^n$, and $f : \mathcal{X} \to [0,1]$ be a fixed predictor for a binary response $Y \in \{0,1\}$ on an input $X \in \mathcal{X}$ such that $(X, Y) \sim \mathcal{D}$. Given a sample $X = x$, $X \sim \mathcal{D}_{\mathcal{X}}$, a feature $j \in [n]$, and a subset $C \subseteq [n] \setminus \{j\}$, define $\Gamma_{j,C} \coloneqq f(\widetilde{X}_{C \cup \{j\}}) - f(\widetilde{X}_C)$ such that $p_{j,C}^{S\text{-}XRT} = \mathbb{E}[\hat{p}_{j,C}^{S\text{-}XRT}]$ and $\gamma_{j,C}^{SHAP} = \mathbb{E}[\Gamma_{j,C}]$. Then,*

$$0 \le p_{j,C}^{S\text{-}XRT} \le 1 - \frac{K}{K+1} \gamma_{j,C}^{SHAP} \tag{11}$$

*and furthermore, if $\gamma_{j,C}^{SHAP} < 0$*

$$\frac{1}{K+1} \left(1 + K(\gamma_{j,C}^{SHAP})^2\right) \le p_{j,C}^{S\text{-}XRT} \le 1, \tag{12}$$

*where $K \ge 1$ is the number of draws of null statistic in the SHAP-XRT procedure.*

We defer the proof of this result to Appendix A.2. We note that the theorem above is presented in expectation, and an analogous finite-sample result can be derived by means of concentration inequalities by estimating $\gamma_{j,C}^{\text{SHAP}}$ with an empirical mean (as long as it is computed over samples independent of $f$). We now discuss the meaning of Theorem 2 and its behavior as a function of $K$ and $L$ in the SHAP-XRT procedure, i.e. the number of draws of null statistic and the number of random reference values sampled at each iteration, respectively.

**Behavior of Equations (11) and (12) as a function of $K$.** As $K \to \infty$, i.e. as the number of draws of null statistic in the SHAP-XRT procedure goes to infinity, $K/(K+1) \to 1$, and Equations (11) and (12) become

$$0 \le p_{j,C}^{\text{S-XRT}} \le 1 - \gamma_{j,C}^{\text{SHAP}} \qquad \text{and} \qquad (\gamma_{j,C}^{\text{SHAP}})^2 \le p_{j,C}^{\text{S-XRT}} \le 1, \tag{13}$$

respectively. Furthermore, we remark that the coefficient $K/(K+1)$ in the bounds above is a mild requirement in practical scenarios. In particular, in order to expect to reject $H_{0,j,C}^{\text{S-XRT}}$ at level $\alpha$ for $\gamma_{j,C}^{\text{SHAP}} = 1$, it suffices for $K$ to grow as $(1 - \alpha)/\alpha$ (e.g., $\alpha = 0.05 \implies K \ge 19$).

**Behavior of $\hat{p}^{\text{S-XRT}}$ as a function of $L$.** Recall that $L$ is the number of samples of reference values over which the test and null statistics are computed in the SHAP-XRT procedure. So far, we presented our results for $L = 1$ and $T$ being the identity. These choices were instrumental for showcasing the connection between the SHAP-XRT test and the Shapley value. Here, we discuss the behavior of the SHAP-XRT $p$-value as $L$ increases. First note that if $L > 1$, the test statistic $T$ cannot simply be the identity because $T$ is a function that maps $\mathbb{R}^L \to \mathbb{R}$. The SHAP-XRT procedure is valid for any choice of test statistic, $T$, and so for $L > 1$, choices of mean, median, quantile, max, etc, are all valid. For example, let $T$ be the mean and in the limit $L \to \infty$, $t = \mathbb{E}[f(\widetilde{X}_{C \cup \{j\}})]$ and $\tilde{t} = \mathbb{E}[f(\widetilde{X}_C)]$. Then, $\hat{p}^{\text{S-XRT}} = (1/(K+1))^{\mathbf{1}[\gamma_{j,C}^{\text{SHAP}} > 0]}$.

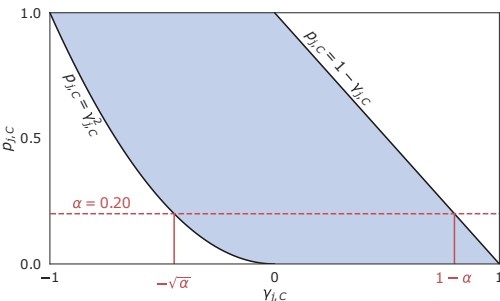

Figure 1: Pictorial representation of the upper and lower bounds on $\hat{p}_{j,C}^{\text{S-XRT}}$ as a function of $\gamma_{j,C}^{\text{SHAP}}$ described in Theorem 2 (as $K \to \infty$). For a critical level $\alpha$ (e.g., $\alpha = 0.20$ in the figure) the bounds indicate under what conditions one can expect to reject, or to fail to reject, the null hypothesis $H_{0,j,C}^{\text{S-XRT}}$ by means of $\gamma_{j,C}^{\text{SHAP}}$.

**Implications of Theorem 2**   The result above provides a novel understanding of the Shapley value from a conditional independence testing perspective. In particular, it shows that each summand $\gamma_{j,C}^{\text{SHAP}}$ (i.e., the expected marginal contribution of feature $j$ to subset $C$) in the Shapley value of feature $j$ provides a lower and an upper bound to $p_{j,C}^{\text{S-XRT}}$, the expected $p$-value of its corresponding SHAP-XRT test (i.e., the test which asks whether the distribution of the response of the model is conditionally independent of $j$ when the features in $C$ are present). Until now, it was unclear whether the marginal contributions carried any statistical meaning. Theorem 2 precisely characterizes under which conditions one can (and cannot) use them to make statistically valid claims about feature importance. In particular, given a critical level $\alpha$ in the SHAP-XRT test, we can identify three separate conditions:

1. *large positive* contributions (i.e., $\gamma_{j,C}^{\text{SHAP}} > 1 - \alpha$) can provide sufficient evidence against the null hypothesis $H_{0,j,C}^{\text{S-XRT}}$, and so one can expect to reject it.

2. *large negative* contributions (i.e., $\gamma_{j,C}^{\text{SHAP}} < -\sqrt{\alpha}$) cannot provide sufficient evidence against the null hypothesis $H_{0,j,C}^{\text{S-XRT}}$, and so one should not expect to be able to reject it.

3. *small* contributions (i.e., $-\sqrt{\alpha} \leq \gamma_{j,C}^{\text{SHAP}} \leq 1 - \alpha$) are noninformative, and so one cannot expect to make any statistically valid claims.

Figure 1 illustrates the bounds (when $K \to \infty$) and the three regimes described above.

### 3.3   The Shapley value as a global hypothesis test

The results presented so far study the statistical meaning of each summand (i.e., contribution) in the Shapley value. However, the overall Shapley value—and not just its individual summands—is used in practice to explain model predictions. Since every summand is linked to the $p$-value of a specific local hypothesis test, a naive approach would be to compare all $p$-values to level $\alpha$, where rejecting at least one hypothesis implies feature $j$ is important with respect to some subset $C \subseteq [n] \setminus \{j\}$. However, this approach is known to inflate the Type I error, and corrections are required (i.e., Bonferroni correction (Shaffer, 1995)). Instead, we ask whether the Shapley value is inherently linked to a *global* hypothesis test.[7] If so, what is the null hypothesis of this test? and what notion of importance does it convey? We now move onto answering these questions, for which we first introduce the definition of a global hypothesis test.

Given $k$ null hypotheses $H_{0,1}, \ldots, H_{0,k}$ with their respective $p$-values $p_1, \ldots, p_k$, the global null hypothesis is defined as $H_0 = \bigcap_{i=1}^{k} H_{0,i}$ and it is true if and only if every individual null hypothesis is true (Hommel, 1983). Several classical procedures exist for global hypothesis testing, both based on the ordering of the $p$-values of the individual tests (Hommel et al., 2011) and on ways of combining them (Tippett et al., 1931; Pearson, 1933; Fisher, 1948). More powerful alternatives have been recently proposed to overcome some of the limitations of these long-established methods, and, importantly, to make no assumptions on the dependency structure of the individual $p$-values (Heard & Rubin-Delanchy, 2018; Futschik et al., 2019; Wilson, 2019; Vovk & Wang, 2020; Liu & Xie, 2020).

---

[7]Here, the term *global* is used according to its meaning in the statistical theory literature, and not in the machine learning explainability literature.

With the above background, we now show how to aggregate all SHAP-XRT tests for feature $j \in [n]$ into a global SHAP-XRT null hypothesis test, provide a valid $p$-value for it, and finally highlight its relation to the Shapley value of feature $j$. Recall that there exists a SHAP-XRT test with null hypothesis $H_{0,j,C}^{\text{S-XRT}}$ for each $C \subseteq [n] \setminus \{j\}$. Using all these tests, we can define the global null hypothesis

$$H_{0,j} = \bigcap_{C \subseteq [n] \setminus \{j\}} H_{0,j,C}^{\text{S-XRT}}, \tag{14}$$

which is false as soon as one of the $H_{0,j,C}^{\text{S-XRT}}$ hypotheses is false. Put into words, rejecting $H_{0,j}$ implies that feature $j$ is important (i.e., it has an effect on the distribution of the response of the model) with respect to at least one subset $C \subseteq [n] \setminus \{j\}$. This way, $H_{0,j}$ tests for a global notion of importance of feature $j$ over all possible conditioning subsets. We remark that this is possible precisely because SHAP-XRT—differently from other conditional independence tests—allows to condition on arbitrary subsets of features. We now provide a valid $p$-value for this global test.

**Lemma 1.** *In the setting of Definition 2, and under the global null hypothesis $H_{0,j}$ (Equation (14)) the random variable*

$$\hat{p}_j = 2 \sum_{C \subseteq [n] \setminus \{j\}} w_C \cdot \hat{p}_{j,C}^{S\text{-}XRT} \tag{15}$$

*is a valid p-value, i.e. $\mathbb{P}[\hat{p}_j \leq \alpha] \leq \alpha, \ \forall \alpha \in [0,1]$.*

This statement follows directly from (Vovk & Wang, 2020, Proposition 9) and its proof is included in Appendix A.3. We remark that the choice of weights is not unique as long as they belong to the simplex. Here, we use the same weights as the Shapley value's (i.e., uniform distribution over all possible subsets), as this will allow us to show that overall Shapley value of feature $j$ indeed provides an upper bound to the expected $p$-value for the global SHAP-XRT test. Alternative weighting functions—which give rise to *semivalues* (Dubey et al., 1981; Wang & Jia, 2023) and *random order values* (Frye et al., 2020b)—are subject of current research (Kwon & Zou, 2022) and we consider the study of their statistical meaning as future work.

**Corollary 1.** *Denote $p_j = \mathbb{E}[\hat{p}_j]$ the expected p-value for the global null $H_{0,j}$. Then,*

$$p_j \leq 2 \left( 1 - \frac{K}{K+1} \phi_j(x, f) \right), \tag{16}$$

*where $\phi_j(x, f)$ is the Shapley value of feature $j \in [n]$.*

The proof is included in Appendix A.4. This result shows that the Shapley value itself, and not only its summands, provides an answer to a statistical test. A straightforward implication of this result is that, given a desired significance level $\alpha$, one can expect to reject $H_{0,j}$ when $\phi_j(x, f) \geq \frac{K+1}{K}(1 - \frac{\alpha}{2})$. In simple terms, a large positive Shapley value, $\phi_j(x, f) \approx 1$, implies that there exists *at least* one $C \subseteq [n] \setminus \{j\}$ for which its expected p-value, $p_{0,j,C}^{\text{S-XRT}}$ is small and $H_{0,j,C}^{\text{S-XRT}}$ is expected to be rejected. This result grants the Shapley value statistical meaning by demonstrating that it is linked to a global hypothesis test.

The natural questions that remain are then: (*i*) when $\phi_j(x, f) \approx 1$ which tests, amongst $\bigcap_{C \subseteq [n] \setminus \{j\}} H_{0,j,C}^{\text{S-XRT}}$, are being rejected? and (*ii*) similarly, do large negative Shapley values, $\phi_j(x, f) \approx -1$, carry any statistical meaning? We answer these questions with the following corollary in a similar fashion to Theorem 2.

**Corollary 2.** *For a feature $j \in [n]$, if $\phi_j(x, f) \geq 1 - \epsilon, \ \epsilon \in (0, 1)$, then $\forall C \subseteq [n] \setminus \{j\}$*

$$p_{j,C}^{S\text{-}XRT} \leq 1 - \frac{K}{K+1} \frac{\widetilde{w} - \epsilon}{\widetilde{w}}, \tag{17}$$

*where $\widetilde{w} = \min_{C \subseteq [n] \setminus \{j\}} w_C$. Furthermore, if $\phi_j(x, f) \leq -1 + \epsilon$ with $\epsilon < \widetilde{w}$, then $\forall C \subseteq [n] \setminus \{j\}$*

$$p_{j,C}^{S\text{-}XRT} \geq \frac{1}{K+1} \left( 1 + K \left( \frac{\epsilon - \widetilde{w}}{\widetilde{w}} \right)^2 \right). \tag{18}$$

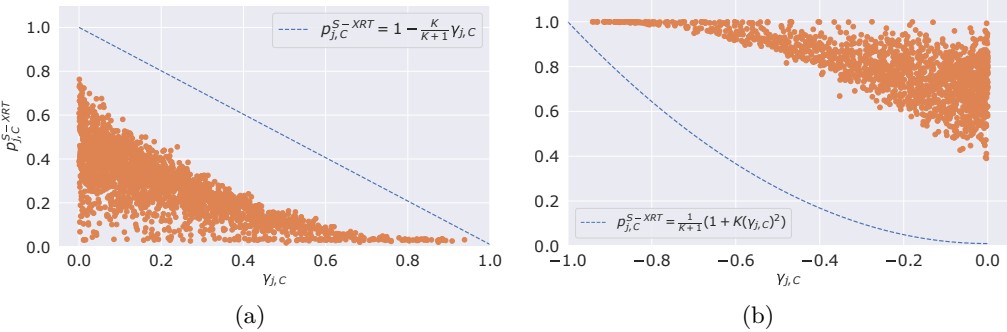

Figure 2: Demonstration of the upper and lower bounds of $p_{j,C}^{\text{S-XRT}}$ for the test $H_{0,j,C}^{\text{S-XRT}} : f(\widetilde{X}_{C \cup \{j\}}) \overset{d}{=} f(\widetilde{X}_C)$ with $j = 1$, $C = \{3, 4, 5, 6\}$ for the known response function experiment. (a) Displays the upper bound with $\theta_2 = 2$ for all samples with $\gamma_{j,C} \geq 0$. (b) Displays the lower bound with $\theta_2 = -2$ for all samples with $\gamma_{j,C} < 0$.

The proof is included in Appendix A.5, and we now provide a few remarks. Firstly, Corollary 2 grants large positive (i.e, $\phi_j(x, f) \approx 1$) and large negative (i.e., $\phi_j(x, f) \approx -1$) Shapley values with statistical meaning. Recall from Corollary 1 that one can expect to reject the global null $H_{0,j}$ for large positive values of $\phi_j(x, f)$. This means that *at least one* of the SHAP-XRT null hypotheses of feature $j$ is expected to be false. In other words, there exists at least one subset $C \subseteq [n] \setminus \{j\}$ such that $j$ affects the distribution of the response of the model when the features in $C$ are present. Corollary 2 makes this statement more precise. In fact, it shows that *all* SHAP-XRT tests needs to have small $p$-values, and one should expect to reject all tests. Symmetrically, it shows that for large negative Shapley values, one cannot expect to reject *any* SHAP-XRT null hypothesis for feature $j$. That is, feature $j$ does not contribute to the distribution of the response of the model given any subset $C \subseteq [n] \setminus \{j\}$.

Secondly, Corollary 2 offers a statistical interpretation of the exponential cost of the Shapley value. Note that for the upper bound in Equation (17) not to be vacuous (i.e., less than 1) and for the lower bound in Equation (18) to hold (i.e., all contributions be negative), $\epsilon$ must be smaller than $\widetilde{w}$ (i.e., $\widetilde{w} - \epsilon > 0 \implies \epsilon < \widetilde{w}$ for Equation (17), and $\epsilon < \widetilde{w}$ for Equation (18) by construction). For $n$ features, the weights $w_C$ in the Shapley value assign a uniform distribution over all possible $2^{n-1}$ subsets $C \subseteq [n] \setminus \{j\}$. It follows that $\widetilde{w}$ is the inverse of the central binomial coefficient, i.e. $\widetilde{w} = \frac{1}{n} \cdot \left( \binom{n-1}{\lfloor \frac{n-1}{2} \rfloor} \right)^{-1} = \mathcal{O}(\frac{\sqrt{n}}{n4^{n/2}})$ by Stirling's approximation. Then, $\epsilon$ needs to decay *exponentially fast with $n$* for the bounds to be informative. This is a well-known limitation of the Shapley value in practical high-dimensional settings. We remark that studying under which settings the computational cost of the Shapley value can be reduced is subject of ongoing research that is orthogonal to the contribution of this paper (Chen et al., 2019; Lundberg et al., 2020; Teneggi et al., 2022a). Rather, Corollary 2 shows the statistical implications of the exponential number of subsets one needs to account for when computing the Shapley value of feature $j$.

Finally, Corollaries 1 and 2 open the door to variations of the Shapley value inspired by more sophisticated and powerful ways of combining $p$-values other than averaging, which may provide more effective in practice while guaranteeing false positive rate control. We consider these potential extensions as part of future work.

## 4    Experiments

We now present three experiments, of increasing complexity, that showcase how the SHAP-XRT procedure can be used in practice to explain machine learning predictions, contextualizing the Shapley value from a statistical viewpoint. All code to reproduce experiments will be made publicly available.

**Known response function**    We first study a case where both the distribution of the data and the ground-truth function are known. Let $d \in \mathbb{N}$ such that $\mathcal{X} \subseteq \mathbb{R}^{2d}$ and denote $X = [X_1, \ldots, X_d] \in \mathbb{R}^{2d}$ the concatenation of $d$ vectors $X_i = [X_{i,1}, X_{i,2}] \in \mathbb{R}^2$. We define the ground-truth function $f(X) = S(\theta^T X)$ where $S$ is the sigmoid function[8] and $\theta = [\theta_1, \theta_2, \cdots, \theta_{2d}] \in \mathbb{R}^{2d}$. For $i \in [d]$, $X_{i,1} \sim \mathcal{N}(1, 1)$ and $X_{i,2} \sim \mathcal{N}(3, 1/4)$ if $X_{i,1} > 3$ and $\mathcal{N}(-1, 1)$ otherwise. We set $d = 3$ and let $\theta = \mathbf{1}_{2d}$ with the exception that $\theta_2 = 2$. We concern ourselves with the null hypothesis $H_{0,j,C}^{\text{S-XRT}} : f(\widetilde{X}_{C \cup \{j\}}) \overset{d}{=} f(\widetilde{X}_C)$ with $j = (1, 2)$ and $C = \{(2, 1), (2, 2), (3, 1), (3, 2)\}$.

---

[8] $S(u) = 1/(1 + e^{-u})$.

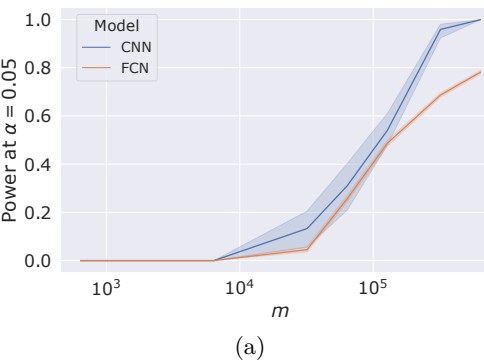 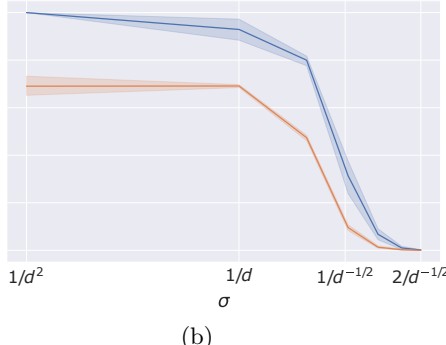

(a)                    (b)

Figure 3: Estimation of the power of $1 - \gamma_{j,C}$ at level 0.05 for a simple CNN and a simple FCN on a synthetic dataset of images. (a) Displays power as a function of the number of samples $m$ used for training, with fixed noise level $\sigma^2 = 1/d^2$, while (b) shows power as a function of the level of the noise in the test dataset while training on $m = 320 \times 10^3$ samples with fixed noise level $\sigma^2 = 1/d^2$. We set $K = 100 \times 10^3$ and $L = 1$ in the SHAP-XRT procedure.

That is, we are aiming to detect when $X_{1,2}$ (i.e., feature $j = (1,2)$) increases the value of $f(\widetilde{X}_{C \cup \{j\}})$ relative to $f(\widetilde{X}_C)$ in the presence of $\{X_2, X_3\}$ (i.e., the features in $C = \{(2,1), (2,2), (3,1), (3,2)\}$). Using the SHAP-XRT procedure with $K = 100$, we calculate $p_{j,C}^{\text{S-XRT}}$ and $\gamma_{j,C}^{\text{SHAP}}$ for 1000 samples. In Figure 2a, for all samples with $\gamma_{j,C}^{\text{SHAP}} \geq 0$, we plot $p_{j,C}^{\text{S-XRT}}$ as a function of $\gamma_{j,C}^{\text{SHAP}}$ and display the upper bound $p_{j,C}^{\text{S-XRT}} = 1 - \frac{K}{K+1}\gamma_{j,C}^{\text{SHAP}}$. To demonstrate the validity of the lower bound, we keep the experiment the same except that now $\theta_2 = -2$. In Figure 2b, for all samples with $\gamma_{j,C}^{\text{SHAP}} < 0$, we plot $p_{j,C}^{\text{S-XRT}}$ as a function of $\gamma_{j,C}^{\text{SHAP}}$ and display the lower bound $p_{j,C}^{\text{S-XRT}} = \frac{1}{K+1}\left(1 + K(\gamma_{j,C}^{\text{SHAP}})^2\right)$. Figure F.3 shows the gap between performing conditional independence testing by means of marginal contributions ($\gamma_{j,C}^{\text{SHAP}}$) or SHAP-XRT $p$-values ($\hat{p}_{j,C}^{\text{S-XRT}}$) as a function of the significance level $\alpha$ by estimating $\mathbb{P}[\gamma_{j,C}^{\text{SHAP}} \geq 1 - \alpha \mid \hat{p}_{j,C}^{\text{S-XRT}} \leq \alpha]$.

**Synthetic image data**  We now present a case where the distribution of the data is known, but we can only estimate the response through some learned model. Let $\mathcal{X} \subseteq \mathbb{R}^{dr \times ds}$, $d, r, s \in \mathbb{N}$ be images composed of an $r \times s$ grid of non-overlapping patches of $d \times d$ pixels, such that $X_{i,j} \in \mathbb{R}^{d \times d}$ is the patch in the $i^{\text{th}}$-row and $j^{\text{th}}$-column. We consider a synthetic dataset of images where the response $Y \in \{0,1\}$ is positive if the input image $X \in \mathbb{R}^{dr \times ds}$ contains at least one instance of a target signal $x_0 \in \mathbb{R}^{d \times d}$. Denote images in $\mathbb{R}^{d \times d}$ as vectors in $\mathbb{R}^D$, $D = d^2$, and let $v \sim \mathcal{N}(0, \sigma^2 \mathbb{I}_D)$ be random noise. Define an *important* distribution $\mathcal{I} = x_0 + v$ for some target signal $x_0 \in \mathbb{R}^D$, and its *unimportant* complement $\mathcal{I}^c = v$, such that $X_{i,j} \sim a_{i,j} \cdot \mathcal{I} + (1 - a_{i,j}) \cdot \mathcal{I}^c$ and $a_{i,j} \sim \text{Bernoulli}(\eta)$ are independent Bernoulli random variables with parameter $\eta$. Then, $Y(X) = 1 \iff \exists (i,j) \in [r] \times [s] : X_{i,j} \sim \mathcal{I}$. In particular, we let $d = 7$, $r = s = 2$ and $x_0$ be a cross (an example is presented in Figure F.1a). Furthermore, we set $\eta = 1 - (1/2)^{1/(r \times s)}$ such that $\mathbb{P}[Y = 0] = \mathbb{P}[Y = 1] = 1/2$. Figure F.1 shows some example images and their respective labels for different noise levels $\sigma^2$. We train a Convolutional Neural Network (CNN) and a Fully Connected Network (FCN) to predict the response $Y$ (see Appendix D.1 for further details). Recalling that the power of a test is defined as $1 - \beta = \mathbb{P}[\text{reject } H_0 \mid H_0 \text{ is false}]$, we estimate the power of performing conditional independence testing via Shapley coefficients at a chosen level $\alpha$ by evaluating $\mathbb{P}[1 - \gamma_{(i,j),C} \leq \alpha \mid H_{0,(i,j),C}^{\text{S-XRT}} \text{ is false}]$ on a test dataset of 320 samples. We remark that $H_{0,(i,j),C}^{\text{S-XRT}}$ is false for all patches $X_{i,j} \sim \mathcal{I}$ such that $\forall (i',j') \in C, X_{(i',j') \in C} \sim \mathcal{I}^c$. Figure 3 shows an estimate of the power of $1 - \gamma_{(i,j),C}$ for both models as a function of number of samples shown during training (Figure 3a), and noise in the test data (Figure 3b) over 5 independent training realizations.

**Real image data**  Finally, we revisit an experiment from Teneggi et al. (2022a) on the BBBC041 dataset (Ljosa et al., 2012), which comprises 1425 images of healthy and infected human blood smears of size $1200 \times 1600$ pixels. The objective of this experiment is to showcase how Theorem 2 translates to a real-world scenario where the same object of interest receives two different Shapley values, and how this affects the $p$-values of the individual SHAP-XRT null hypotheses. Here, the task is to label positively images that contain at least one *trophozoite*—an infectious type of cell. We apply transfer-learning to a ResNet18 (He et al., 2016) pretrained on ImageNet (Deng et al., 2009) (see Appendix D.2 for further details). After training, our model achieves a validation accuracy greater than 99%. Computing the exact Shapley value for each

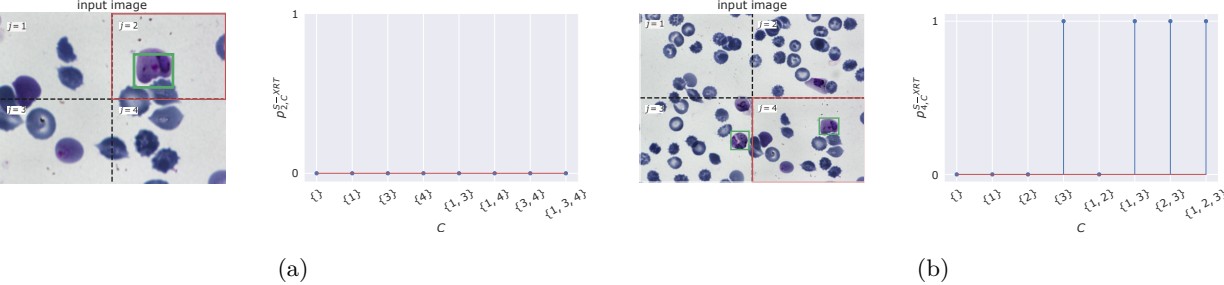

(a)                                     (b)

Figure 4: Demonstration of Shapley value ($\phi_j$) and the $p$-values of their respective hypothesis tests, $p_{j,C}$, for two images of the BBBC041 dataset. Here, each feature is a quadrant, and important regions are those containing trophozoites (indicated by ground-truth green bounding box).

pixel is intractable, hence we take an approach similar to that of h-Shap (Teneggi et al., 2022a) and define features as quadrants in order to compute the $p$-value of each SHAP-XRT null hypothesis. Since there are 4 quadrants, each Shapley value ($\phi_j$) comprises only 8 marginal contributions ($\gamma_{j,C}$) that can be computed exactly without any need for approximation. We extend the original implementation of h-Shap to return the SHAP-XRT tests that reject (or fail to reject) their respective nulls, thus assigning a collection of $p$-values to every quadrant. We note that in this experiment, we use the *unconditional* expectation over the training set to mask features—as it is commonly done in popular Shapley-based explanation methods. Figure F.2 shows the reference value used for masking and some example masked inputs. Figure 4 presents the two examples. The first, Figure 4a, depicts a case where there exists only 1 cell in the upper right quadrant, i.e. $j = 2$. Thus, this quadrant receives a Shapley value $\phi_2 \approx 1$. Naturally, all $p$-values are approximately zero (as also guaranteed by Corollary 2). This implies that the quadrant $j = 2$ is statistically important in the sense that all SHAP-XRT null hypotheses are rejected. On the other hand, in Figure 4b, there are two quadrants that contain sick cells, i.e. $j = 3, 4$, and $\phi_3 \approx \phi_4 \approx 0.5$ by symmetry. Based on our theoretical results, $\phi_3$ and $\phi_4$ can be decomposed in the sum of 8 terms that bound the $p$-values of their respective SHAP-XRT tests. As can be seen, $j = 4$ is indeed statistically important in that half of the null hypotheses are rejected (i.e., all tests $H_{0,4,C}^{\text{S-XRT}}$ such that $3 \notin C$). However, the bound in Corollary 1 shows that a Shapley value of $\phi_4 \approx 0.5$ is not large enough to reject the global null $H_{0,4}$ even if it is clearly false. These results highlight the suboptimality of the Shapley value from a statistical testing perspective, which motivates future work to develop more powerful alternatives. We expand on this experiment in Appendix E.

## 5 Conclusion

The Shapley value and conditional hypothesis testing appear as two unrelated approaches to local (i.e., sample specific) interpretability of machine learning models. In this work, we have shown that the two are tightly connected in that the former involves the computation of specific conditional hypothesis tests, and that every summand in the Shapley value can be used to bound the $p$-values of such tests. For the first time, this perspective grants the Shapley value with a precise statistical meaning. We presented numerical experiments on synthetic and real data of increasing complexity to depict our theoretical results in practice. We hope that this work will enable the further and precise understanding of the meaning of (very popular) game theoretic quantities in the context of statistical learning. Restricting the Shapley value to an *a-priori* subset of null hypotheses may prove successful in devising useful algorithms in certain scenarios, and alternative ways other than the weighted arithmetic mean—which is employed by the Shapley value—may yield powerful procedures to combine SHAP-XRT's $p$-values to test for global null hypotheses.

### Acknowledgments

The authors thank the anonymous reviewers who helped improve the clarity and presentation of this manuscript. This research was supported by the NSF CAREER Award CCF 2239787. Y. R. was supported by the Israel Science Foundation (grant No. 729/21). Y. R. also thanks the Career Advancement Fellowship, Technion, for providing research support.

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

# A   Proofs

We briefly summarize the notation used in this section. Recall that $f : \mathbb{R}^n \to [0, 1]$ is a fixed predictor for a binary response $Y \in \{0, 1\}$ on some input $X \in \mathbb{R}^n$ for an unknown distribution $\mathcal{D}$ such that $(X, Y) \sim \mathcal{D}$. For a sample $X = x$, $X \sim \mathcal{D}_\mathcal{X}$ and each subset $C \subseteq [n]$ denote $\widetilde{X}_C = [x_C, X_{-C}] \in \mathbb{R}^n$ the random vector that agrees with $x$ in the features in $C$ and that takes a reference value sampled from its conditional distribution $X_{-C} \mid X_C = x_C$ in its complement $-C := [n] \setminus C$. For brevity, we write $\widetilde{X}_C \sim \mathcal{D}_{X_C=x_C}$. Finally, for a feature $j \in [n]$, and a subset $C \subseteq [n] \setminus \{j\}$ denote

$$\Gamma_{j,C} := f(\widetilde{X}_{C \cup \{j\}}) - f(\widetilde{X}_C) \tag{19}$$

the random marginal contribution of feature $j$ to subset $C$ such that

$$p_{j,C}^{\text{S-XRT}} := \mathbb{E}[\hat{p}_{j,C}^{\text{S-XRT}}] \quad \text{and} \quad \gamma_{j,C}^{\text{SHAP}} := \mathbb{E}[\Gamma_{j,C}] \tag{20}$$

are the expected $p$-value of the SHAP-XRT procedure and the expected marginal contribution, respectively.

## A.1   Proof of Theorem 1

Here, we show that the $p$-value returned by the SHAP-XRT procedure is valid, i.e. under $H_{0,j,C}^{\text{S-XRT}}$, $\mathbb{P}[\hat{p}_{j,C}^{\text{S-XRT}} \leq \alpha] \leq \alpha, \ \forall \alpha \in [0, 1]$.

*Proof.* Recall that given a sample $X = x$, $X \sim \mathcal{D}_\mathcal{X}$, a feature $j \in [n]$, and a subset $C \subseteq [n] \setminus \{j\}$, the null hypothesis of the test is

$$H_{0,j,C}^{\text{S-XRT}} : \ f(\widetilde{X}_{C \cup \{j\}}) \stackrel{d}{=} f(\widetilde{X}_C). \tag{21}$$

Denote $\widetilde{\boldsymbol{X}}_{C \cup \{j\}}, \widetilde{\boldsymbol{X}}_C$ the random matrices containing $L \geq 1$ duplicates of $\widetilde{X}_{C \cup \{j\}}$ and $\widetilde{X}_C$, respectively, such that $\widetilde{Y}_{C \cup \{j\}} = f(\widetilde{\boldsymbol{X}}_{C \cup \{j\}})$ and $\widetilde{Y}_C = f(\widetilde{\boldsymbol{X}}_C)$ are the predictions of the model. By definition, under $H_{0,j,C}^{\text{S-XRT}}$

$$\widetilde{Y}_{C \cup \{j\}}, \widetilde{Y}_C^{(1)}, \ldots, \widetilde{Y}_C^{(K)}, \tag{22}$$

$K \geq 1$, are i.i.d. hence exchangeable. It follows that for any choice of test statistic $T$, the random variables $T(\widetilde{Y}_{C \cup \{j\}}), T(\widetilde{Y}_C^{(1)}), \ldots, T(\widetilde{Y}_C^{(K)})$ are also exchangeable. We conclude that $\mathbb{P}[\hat{p}_{j,C}^{\text{S-XRT}} \leq \alpha] \leq \alpha, \ \forall \alpha \in [0, 1]$. $\square$

## A.2   Proof of Theorem 2

Here, we prove the bounds on the expected $p$-value of the SHAP-XRT procedure presented in Theorem 2.

### A.2.1   Useful inequalities

We start by including some known inequalities that will become useful in the proof of our theorem.

**Proposition A.1** (Paley-Zygmund's inequality Paley & Zygmund (1932))**.** *If $Z$ is a nonnegative random variable with finite variance, and $0 \leq \theta \leq 1$, then*

$$\mathbb{P}[Z > \theta \mathbb{E}[Z]] \geq \frac{(1 - \theta)^2 \mathbb{E}[Z]^2}{Var(Z) + (1 - \theta)^2 \mathbb{E}[Z]^2}. \tag{23}$$

**Proposition A.2** (Bhatia-Davis' inequality Bhatia & Davis (2000))**.** *If $Z$ is a bounded random variable in $[m, M]$, then*

$$Var(Z) \leq (M - \mathbb{E}[Z])(\mathbb{E}[Z] - m) \tag{24}$$

### A.2.2 Lemmas

We present two lemmas that provide upper and lower bounds to the negative tail of $\Gamma_{j,C}$, respectively. Note that $\Gamma_{j,C} \in [-1, 1]$ because $f \in [0, 1]$. Herein, we define the nonnegative random variable $Z := 1 - \Gamma_{j,C}$ such that

$$\mathbb{E}[Z] = 1 - \gamma_{j,C}^{\mathrm{SHAP}} \qquad \text{and} \qquad \mathbb{P}[\Gamma_{j,C} \leq 0] = \mathbb{P}[Z \geq 1]. \tag{25}$$

**Lemma A.1** (Upper bound)**.** *By Markov's inequality, it holds that*

$$\mathbb{P}[\Gamma_{j,C} \leq 0] \leq 1 - \gamma_{j,C}^{SHAP}. \tag{26}$$

*Proof.* Applying Markov's inequality to $Z$ directly yields

$$\mathbb{P}[\Gamma_{j,C} \leq 0] = \mathbb{P}[Z \geq 1] \leq \mathbb{E}[Z] = 1 - \gamma_{j,C}^{\mathrm{SHAP}}. \tag{27}$$

$\square$

**Lemma A.2** (Lower bound)**.** *If $\gamma_{j,C}^{SHAP} < 0$, by the Paley-Zygmund's and Bhatia-Davis' inequalities, it holds that*

$$\mathbb{P}[\Gamma_{j,C} \leq 0] \geq (\gamma_{j,C}^{SHAP})^2. \tag{28}$$

*Proof.* Note that $\gamma_{j,C}^{\mathrm{SHAP}} < 0 \implies 1/\mathbb{E}[Z] < 1$ and let $\theta = 1/\mathbb{E}[Z]$. Then, the Paley-Zygmund's inequality yields

$$\mathbb{P}[\Gamma_{j,C} \leq 0] \geq \frac{(\gamma_{j,C}^{\mathrm{SHAP}})^2}{\mathrm{Var}(1 - \Gamma_{j,C}) + (\gamma_{j,C}^{\mathrm{SHAP}})^2}. \tag{29}$$

Since $1 - \Gamma_{j,C} \in [0, 2]$, the Bhatia-Davis' inequality implies

$$\mathrm{Var}(1 - \Gamma_{j,C}) \leq (2 - \mathbb{E}[1 - \Gamma_{j,C}])(\mathbb{E}[1 - \Gamma_{j,C}] - 0) \tag{30}$$
$$= (1 + \gamma_{j,C}^{\mathrm{SHAP}})(1 - \gamma_{j,C}^{\mathrm{SHAP}}) \tag{31}$$
$$= 1 - (\gamma_{j,C}^{\mathrm{SHAP}})^2. \tag{32}$$

Therefore,

$$\mathbb{P}[\Gamma_{j,C} \leq 0] \geq \frac{(\gamma_{j,C}^{\mathrm{SHAP}})^2}{\mathrm{Var}(1 - \Gamma_{j,C}) + (\gamma_{j,C}^{\mathrm{SHAP}})^2} \tag{33}$$

$$\geq \frac{(\gamma_{j,C}^{\mathrm{SHAP}})^2}{1 - (\gamma_{j,C}^{\mathrm{SHAP}})^2 + (\gamma_{j,C}^{\mathrm{SHAP}})^2} = (\gamma_{j,C}^{\mathrm{SHAP}})^2. \tag{34}$$

$\square$

### A.2.3 Proof of Equation (11)

Here, we show that

$$0 \leq p_{j,C}^{\mathrm{S\text{-}XRT}} \leq 1 - \frac{K}{K+1} \gamma_{j,C}^{SHAP}, \tag{35}$$

where $K \geq 1$ is the number of draws of null statistic in the SHAP-XRT procedure.

*Proof.* Recall that in the setting of Theorem 2, the test statistic $T$ is the identity and $L = 1$ in the SHAP-XRT procedure. Then,

$$p_{j,C}^{\text{S-XRT}} = \mathbb{E}[\hat{p}_{j,C}^{\text{S-XRT}}] = \mathbb{E}\left[\frac{1}{K+1}\left(1 + \sum_{k=1}^{K}\mathbf{1}[\tilde{t}^{(k)} \geq t]\right)\right] \tag{36}$$

$$= \frac{1}{K+1}\left(1 + \sum_{k=1}^{K}\mathbb{E}[\mathbf{1}[\tilde{t}^{(k)} \geq t]]\right) \tag{37}$$

$$= \frac{1}{K+1}\left(1 + K\mathbb{P}[\tilde{t}^{(k)} \geq t]\right) \tag{38}$$

$$= \frac{1}{K+1}\left(1 + K\mathbb{P}[\Gamma_{j,C} \leq 0]\right) \qquad (t = f(\widetilde{X}_{C \cup \{j\}}),\ \tilde{t}^{(k)} = f(\widetilde{X}_C^{(k)})) \tag{39}$$

$$\leq \frac{1}{K+1}\left(1 + K(1 - \gamma_{j,C}^{\text{SHAP}})\right) \qquad \text{(from Lemma A.1)} \tag{40}$$

$$= 1 - \frac{K}{K+1}\gamma_{j,C}^{\text{SHAP}} \tag{41}$$

which is the desired result. $\square$

### A.2.4 Proof of Equation (12)

Here, we show that if $\gamma_{j,C}^{\text{SHAP}} < 0$

$$\frac{1}{K+1}\left(1 + K(\gamma_{j,C}^{\text{SHAP}})^2\right) \leq p_{j,C}^{\text{S-XRT}} \leq 1. \tag{42}$$

where $K \geq 1$ is the number of draws of null statistic in the SHAP-XRT procedure.

*Proof.* Recall that in the setting of Theorem 2, the test statistic $T$ is the identity and $L = 1$ in the SHAP-XRT procedure. Then, similar to above

$$p_{j,C}^{\text{S-XRT}} = \mathbb{E}[\hat{p}_{j,C}^{\text{S-XRT}}] = \frac{1}{K+1}\left(1 + K\mathbb{P}[\Gamma_{j,C} \leq 0]\right) \tag{43}$$

$$\geq \frac{1}{K+1}\left(1 + K(\gamma_{j,C}^{\text{SHAP}})^2\right) \qquad \text{(from Lemma A.2)} \tag{44}$$

which is the desired result. $\square$

### A.3 Proof of Lemma 1

Here, we prove that

$$\hat{p}_j = 2 \sum_{C \subseteq [n]\setminus\{j\}} w_C \cdot \hat{p}_{j,C}^{\text{S-XRT}} \tag{45}$$

is a valid $p$-value for $H_{0,j} = \bigcap_{C \subseteq [n]\setminus\{j\}} H_{0,j,C}^{\text{S-XRT}}$.

*Proof.* The proof follows directly from (Vovk & Wang, 2020, Proposition 9). Note that the weights $w_C = 1/n \cdot \binom{n-1}{|C|}^{-1}$ belong to $\Delta^{2^n - 1}$ (i.e., $w_C > 0$, $\sum_{C \subseteq [n]\setminus\{j\}} w_C = 1$). Furthermore, note that $1/w := 1/\max_{C \subseteq [n]\setminus\{j\}} w_C = n$. Then, when $n \geq 2$ (i.e., for at least 2 features), the result follows by setting $r = 1$ in (Vovk & Wang, 2020, Proposition 9). $\square$

### A.4 Proof of Corollary 1

Here, we prove that the Shapley value of feature $j$ can be used to test for a global null hypothesis

$$H_{0,j} = \bigcap_{C \subseteq [n]\setminus\{j\}} H_{0,j,C}^{\text{S-XRT}}. \tag{46}$$

*Proof.* Given feature $j \in [n]$, recall from Lemma 1 that

$$p_j = \mathbb{E}[\hat{p}_j] = 2 \sum_{C \subseteq [n] \setminus \{j\}} w_C \cdot p_{j,C}^{\text{S-XRT}} \tag{47}$$

is the expected $p$-value for the global null $H_{0,j}$. Then,

$$\phi_j(x, f) = \sum_{C \subseteq [n] \setminus \{j\}} w_C \cdot \gamma_{j,C}^{\text{SHAP}} \tag{48}$$

$$\leq \frac{K+1}{K} \sum_{C \subseteq [n] \setminus \{j\}} w_C (1 - p_{j,C}^{\text{S-XRT}}) \qquad \text{(from Theorem 2)} \tag{49}$$

$$= \frac{K+1}{K} \left( 1 - \sum_{C \subseteq [n] \setminus \{j\}} w_C \cdot p_{j,C}^{\text{S-XRT}} \right) \tag{50}$$

$$= \frac{K+1}{K} \left( 1 - \frac{p_j}{2} \right), \tag{51}$$

from which it follows

$$p_j \leq 2 \left( 1 - \frac{K}{K+1} \phi_j(x, f) \right), \tag{52}$$

which is the desired result. □

### A.5   Proof of Corollary 2

Here, we prove the bounds on the expected $p$-value of the global SHAP-XRT null hypothesis presented in Corollary 2.

### A.5.1   Proof of Equation (17)

Here, we show that if $\phi_j(x, f) \geq 1 - \epsilon, \ \epsilon \in (0, 1)$, then $\forall C \subseteq [n] \setminus \{j\}$

$$p_{j,C}^{\text{S-XRT}} \leq 1 - \frac{K}{K+1} \frac{\widetilde{w} - \epsilon}{\widetilde{w}}, \tag{53}$$

where $\widetilde{w} = \min_{C \subseteq [n] \setminus \{j\}} w_C$.

*Proof.* Suppose $\phi_j(x, f) \geq 1 - \epsilon, \ \epsilon \in (0, 1)$ and fix $C^* \in [n] \setminus \{j\}$ such that

$$\phi_j(x, f) = \sum_{C \subseteq [n] \setminus \{j\}} w_C \cdot \gamma_{j,C}^{\text{SHAP}} = w_{C^*} \cdot \gamma_{j,C^*}^{\text{SHAP}} + \sum_{C \neq C^*} w_C \cdot \gamma_{j,C}^{\text{SHAP}} \geq 1 - \epsilon, \tag{54}$$

which implies

$$\gamma_{j,C^*}^{\text{SHAP}} \geq \frac{1 - \epsilon - \sum_{C \neq C^*} w_C \cdot \gamma_{j,C}^{\text{SHAP}}}{w_{C^*}} \tag{55}$$

$$\geq \frac{1 - \epsilon - \sum_{C \neq C^*} w_C}{w_{C^*}} \qquad (\max \gamma_{j,C}^{\text{SHAP}} = 1) \tag{56}$$

$$= \frac{1 - \epsilon - (1 - w_{C^*})}{w_{C^*}} \qquad \left( \sum_{C \subseteq [n] \setminus \{j\}} w_C = 1 \right) \tag{57}$$

$$= \frac{w_{C^*} - \epsilon}{w_{C^*}} \geq \frac{\widetilde{w} - \epsilon}{\widetilde{w}}, \ \widetilde{w} = \min_{C \subseteq [n] \setminus \{j\}} w_C. \tag{58}$$

Note that the last inequality follows from $(w_{C^*} - \epsilon)/w_{C^*}$ being an increasing function of $w_{C^*}$ for $\epsilon > 0$. Then,

$$\min_{C \subseteq [n] \setminus \{j\}} \gamma_{j,C}^{\text{SHAP}} \geq \frac{\widetilde{w} - \epsilon}{\widetilde{w}} \implies p_{j,C}^{\text{S-XRT}} \leq 1 - \frac{K}{K+1} \frac{\widetilde{w} - \epsilon}{\widetilde{w}}. \qquad \text{(from Theorem 2)} \tag{59}$$

□

### A.5.2 Proof of Equation (18)

Here, we show that if $\phi_j(x, f) \leq -1 + \epsilon$ with $\epsilon < \widetilde{w}$, $\widetilde{w} = \min_{C \subseteq [n] \setminus \{j\}} w_C$, then $\forall C \subseteq [n] \setminus \{j\}$

$$p_{j,C}^{\text{S-XRT}} \geq \frac{1}{K+1} \left( 1 + K \left( \frac{\epsilon - \widetilde{w}}{\widetilde{w}} \right)^2 \right) \tag{60}$$

*Proof.* Similarly to above, suppose $\phi_j(x, f) \leq -1 + \epsilon$, $\epsilon \in (0, 1)$ and fix $C^* \in [n] \setminus \{j\}$ such that

$$\phi_j(x, f) = \sum_{C \subseteq [n] \setminus \{j\}} w_C \cdot \gamma_{j,C}^{\text{SHAP}} = w_{C^*} \cdot \gamma_{j,C^*}^{\text{SHAP}} + \sum_{C \neq C^*} w_C \cdot \gamma_{j,C}^{\text{SHAP}} \leq -1 + \epsilon, \tag{61}$$

which implies

$$\gamma_{j,C^*}^{\text{SHAP}} \leq \frac{\epsilon - \widetilde{w}}{\widetilde{w}} \tag{62}$$

and $\gamma_{j,C^*}^{\text{SHAP}} < 0 \iff \epsilon < \widetilde{w}$. Then,

$$\max_{C \subseteq [n] \setminus \{j\}} \gamma_{j,C}^{\text{SHAP}} \leq \frac{\epsilon - \widetilde{w}}{\widetilde{w}} < 0 \implies p_{j,C}^{\text{S-XRT}} \geq \frac{1}{K+1} \left( 1 + K \left( \frac{\epsilon - \widetilde{w}}{\widetilde{w}} \right)^2 \right). \quad \text{(from Theorem 2)} \tag{63}$$

$\square$

## B  Axioms of the Shapley value

Recall that the tuple $([n], v)$, $[n] := \{1, \ldots, n\}$, $v : \mathcal{P}(n) \to \mathbb{R}^+$ is an $n$-person TU game with characteristic function $v$, such that $\forall C \subseteq [n]$, $v(C)$ is the score accumulated by the players in the coalition $C$. Then, the Shapley values $\phi_1([n], v), \ldots, \phi_n([n], v)$ of the game $([n], v)$ (see Definition 1) are the only solution concept that satisfies the following axioms (Shapley, 1951):

**Axiom 1** (Additivity). The Shapley values sum up to the utility accumulated when all players participate in the game (i.e. the *grand coalition* of the game)

$$\sum_{j=1}^{n} \phi_j([n], v) = v([n]). \tag{64}$$

**Axiom 2** (Nullity). If a player does not contribute to any coalition, its Shapley value is 0

$$\forall C \subseteq [n] \setminus \{j\}, \ v(C \cup \{j\}) = v(C) \implies \phi_j([n], v) = 0. \tag{65}$$

**Axiom 3** (Symmetry). If the contributions of two players to any coalition are the same, their Shapley values are the same

$$\forall C \subseteq [n] \setminus \{j, k\}, \ v(C \cup \{j\}) = v(C \cup \{k\}) \implies \phi_j([n], v) = \phi_k([n], v). \tag{66}$$

**Axiom 4** (Linearity). Given $([n], v)$, $([m], v)$, the Shapley value of the union of the two games (i.e. $\phi_j([n] \cup [m], v)$) is equal to the sum of the Shapley values of the individual games (i.e. $\phi_j([n], v)$ and $\phi_j([m], v)$, respectively)

$$\phi_j([n] \cup [m], v) = \phi_j([n], v) + \phi_j([m], v). \tag{67}$$

Finally, we note that Axioms 2–4 can be replaced by a fifth one, usually referred to as *balanced contribution* (Fryer et al., 2021), although this is not necessary to derive the definition of the Shapley value.

## C  Algorithms

Algorithm C.1 summarizes the CRT procedure by Candes et al. (2018).

---

**Algorithm C.1** Conditional Randomization Test

---

**procedure** CRT(data $X = (x^{(1)}, \ldots, x^{(m)}) \in \mathbb{R}^{m \times n}$, response $Y = (y^{(1)}, \ldots, y^{(m)}) \in \mathbb{R}^m$, feature $j \in [n]$,
test statistic $T$, number of null draws $K \in \mathbb{N}$)

    Compute the test statistic, $t \leftarrow T(X_j, Y, X_{-j})$

    **for** $k \leftarrow 1, \ldots, K$ **do**

        Sample $X_j^{(i)} \sim X_j \mid X_{-j} = x_{-j}^{(i)}, \ i = 1, \ldots, m$

        $X_j^{(k)} \leftarrow (X_j^{(1)}, \ldots, X_j^{(m)})$

        Compute the null statistic, $\tilde{t}^{(k)} \leftarrow T(X_j^{(k)}, X_{-j}, Y)$

    **end for**

    **return** A (one-sided) $p$-value

$$\hat{p}_j^{\text{CRT}} = \frac{1}{K+1} \left( 1 + \sum_{k=1}^{K} \mathbf{1}[\tilde{t}^{(k)} \geq t] \right)$$

**end procedure**

---

# D   Experimental details

Before describing the experimental details, we note that all experiments were run on an NVIDIA Quadro RTX 5000 with 16 GB of RAM memory on a private server with 96 CPU cores. All scripts were run on PyTorch `1.11.0`, Python `3.8.13`, and CUDA `10.2`.

## D.1   Synthetic image data

Here, we describe the model architectures and the training details for the synthetic image datasets. Recall that $\mathcal{X} \subseteq \mathbb{R}^{dr \times ds}$, $d, r, s \in \mathbb{N}$ are images composed of an $r \times s$ grid of non-overlapping patches of $d \times d$ pixels, such that $X_{i,j} \in \mathbb{R}^{d \times d}$ is the patch in the $i^{\text{th}}$-row and $j^{\text{th}}$-column.

We train a CNN with one filter with stride $d$, and a two-layer FCN with ReLU activation. In particular:

$$f^{\text{CNN}}(X) = S \left( b_0 + \sum_{i,j \in [r] \times [s]} \langle W_0, X_{i,j} \rangle \right), \tag{68}$$

and

$$f^{\text{FCN}}(X) = S \left( b_1 + \langle W_1, \text{ReLU}\left( b_0 + \langle W_0, X \rangle \right) \right), \tag{69}$$

where $S(u) = 1/(1 + e^{-u})$ is the sigmoid function, and $\text{ReLU}(u) = [0, x]_+$ is the rectified linear unit (Nair & Hinton, 2010). We train both models for one epoch on $m$ i.i.d. samples and a batch size of 64. We note that we use Adam (Kingma & Ba, 2014) with learning rate of 0.001, and SGD with learning rate of 0.01 for $f^{\text{CNN}}$ and $f^{\text{FCN}}$, respectively, to achieve optimal validation accuracy.

## D.2   Real image data

Here, we present the details of the training process for the experiment on the BBBC041 dataset (Ljosa et al., 2012) (which is publicly available at `https://bbbc.broadinstitute.org/BBBC041`). Recall that the dataset comprises 1425 images of healthy and infected human blood smears. We split the original dataset into a training and validation split using an 80/20 ratio, respectively. This way, we train our model on 589 positive and 608 negative images, and validate on 112 positive and 116 negative images.

We apply transfer learning to a ResNet18 (He et al., 2016) pretrained on ImageNet (Deng et al., 2009). We optimize all parameters of the network for 25 epochs using binary-cross entropy loss and Adam optimizer, with a learning rate of 0.0001 and learning rate decay of 0.2 every 10 epochs. At training time, we augment the dataset with random horizontal flips.

# E  Supplementary results for real image data experiment

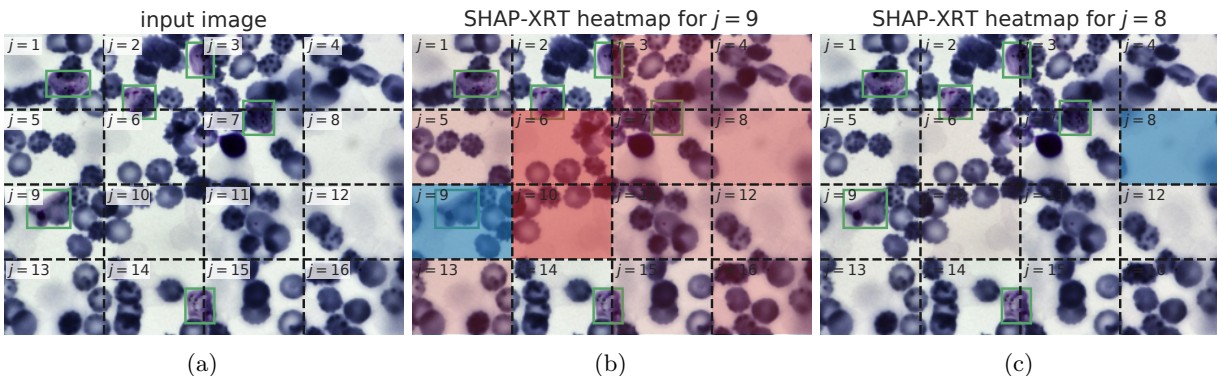

| input image | SHAP-XRT heatmap for $j = 9$ | SHAP-XRT heatmap for $j = 8$ |
|:---:|:---:|:---:|
| (a) | (b) | (c) |

Here, we revisit the real image data experiment presented in Section 4. Instead of using 4 quadrants, we further subdivide them to obtain 16 features, $j = 1, \ldots, 16$, as displayed in Figure E.1a. The input image contains 6 trophozoites that are highlighted with green bounding boxes. For each feature, we compute the $p$-values of all $2^{15}$ SHAP-XRT null hypotheses. Figures E.1b and E.1c show the SHAP-XRT heatmaps for features $j = 9$ (which contains a trophozoite) and $j = 8$ (which does not contain a trophozoite), respectively. In the heatmaps, each feature $j' \in [16] \setminus \{j\}$ is colored accordingly to the number of SHAP-XRT null hypotheses $H_{j,C}^{\text{S-XRT}}$, $j' \in C$ that are rejected. That is, the higher number of tests are rejected such that $j' \in C$, the stronger the color of feature $j'$. Figure E.1b shows that for an important feature (i.e., a feature that does contain a trophozoite), the heatmap concentrates on features that do not contain any trophozoites. This agrees with intuition that when $C$ does not contain any trophozoites, adding a feature with a trophozoite will affect the distribution of the output of the model. Symmetrically, when $C$ already contains some trophozoites, adding another will not affect the distribution. Figure E.1c shoes that for an unimportant feature (i.e., a feature that does not contain a trophozoite), the heatmap is empty because no matter whether $C$ contains a trophozoite, adding an empty feature will not affect the distribution of the response.

## F Figures

Here we include supplementary figures.

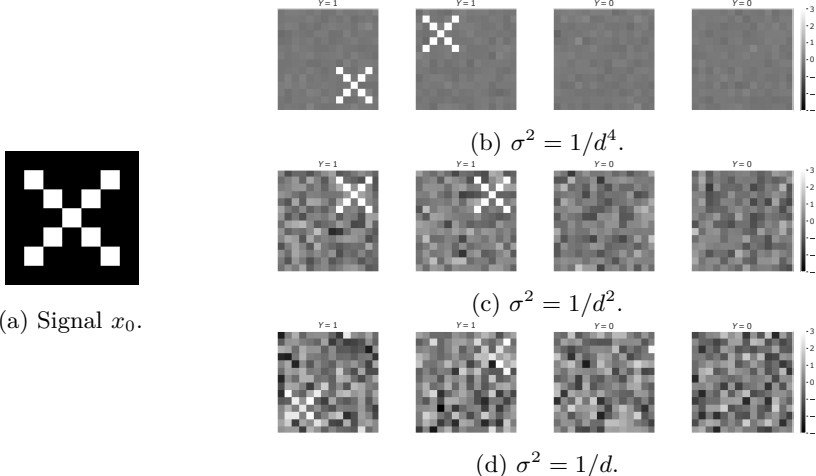

(a) Signal $x_0$.

(b) $\sigma^2 = 1/d^4$.

(c) $\sigma^2 = 1/d^2$.

(d) $\sigma^2 = 1/d$.

Figure F.1: (a) Target signal $x_0$: a cross of size $d \times d$ pixels, $d = 7$. (b), (c), (d) Examples of positive and negative images with increasing levels of noise, $\sigma^2 = 1/d^4$, $1/d^2$, and $1/d$, respectively. Note that all images are normalized to have zero-mean and unit variance.

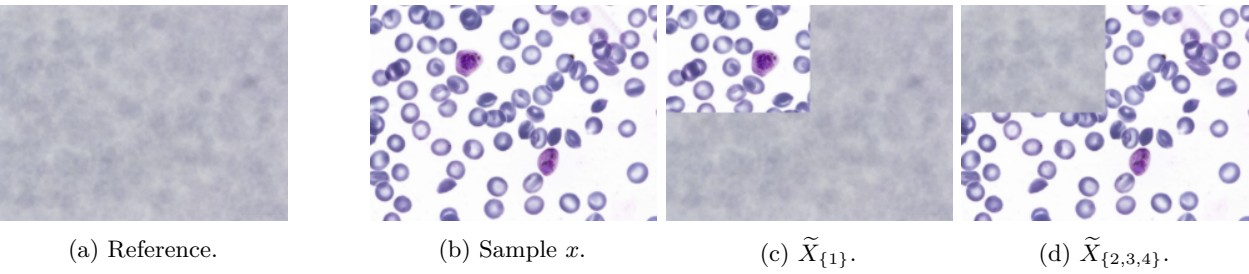

(a) Reference.      (b) Sample $x$.      (c) $\widetilde{X}_{\{1\}}$.      (d) $\widetilde{X}_{\{2,3,4\}}$.

Figure F.2: Some examples of masked inputs from the BBBC041 dataset. (a) Shows the unimportant reference (i.e. the unconditional expectation over the training split) used to mask features. We verify that the reference image does not contain any signal by checking that the output of $f$ is $\approx 0$. (b) Shows the original sample $x$; (c) and (d) show some masked versions of $x$ when conditioning on the sets $C = \{1\}$ and $C = \{2, 3, 4\}$, respectively.

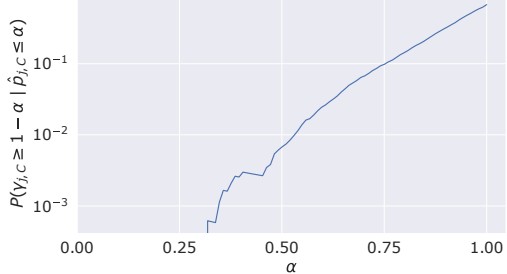

Figure F.3: Estimation of $\mathbb{P}[\gamma_{j,C}^{\text{SHAP}} \geq 1 - \alpha \mid \hat{p}_{j,C}^{\text{S-XRT}} \leq \alpha]$ as a function of significance level $\alpha$ for $j = 2$, $C = \{(2, 1), (2, 2), (3, 1), (3, 2)\}$ in the known response experiment. This evaluates the gap between performing conditional independence testing by means of marginal contributions or the SHAP-XRT $p$-values. Estimates were computed over 100 independent draws of the SHAP-XRT $p$-values for a fixed dataset of $N = 5,000$ data points.

