# OpenReview forum: "SHAP-XRT: The Shapley Value Meets Conditional Independence Testing"
_TMLR — Accepted by TMLR_

### Review · Reviewer_1xWV · 2023-07-24

**Summary Of Contributions:**

The paper aims to give statistical interpretations for Shapley values in the context of machine learning interpretability.  Here, the Shapley value of a feature given a data point is the expected contribution of the target feature to the label, averaged over a reference set of other features.  The authors first present a test for conditional independence of individual features, which is a local (i.e., fixing a data point) variant of standard tests used in ML interpretability.  The idea of the test is: fixing a data point and given a (target) feature and a set of other (reference) features, we want to see whether conditioned on the reference feature set, the label is independent of the target feature (which would mean the target feature is unimportant).  The main results are that (1) each individual term in the Shapley value corresponding to a feature bounds the p-value of the corresponding conditional independence test for the same feature, and (2) the Shapley value itself bounds the p-value of the "joint" test, where the null hypothesis is the intersection of the null hypotheses of all tests for the same feature with different reference sets.  The authors discuss how these bounds can be used in interpretability, and conduct experiments to support these claims.

**Audience:**

Yes

**Claims And Evidence:**

Yes

**Requested Changes:**

(also putting some detailed comments here; authors can use their own judgment)

Introduction, medicine example with 0.30, 0.15, ...: I'm not sure I fully understand this example.  Without thinking too much I thought all 4 should be reported?  Or maybe the point is weight and height are correlated and the true causally important feature is the patient's weight?  In any case it might help to briefly explain here.

First sentence in sec 2.1, "game theory (Owen, 2013)": maybe something like "see, e.g., (Owen, 2013)" would sound better?  Also it seems like you are talking exclusively about *cooperative* game theory, so might as well cite something more specific.

Eq (3): I guess \phi_j can be negative here (whereas in coop game theory it's normally nonnegative)?  It might help to briefly comment on it here (e.g., intuitively an important feature has a Shapley value whose *absolute value* is large).

Sentence after eq (3): I guess there has to be some constraints on C^* (otherwise one would simply choose [n])?

Sec 3.1: it's good to be formal and precise, but I'd also say upfront how this is connected to CRT, which I imagine would be appreciated by experts.

General criticism on thm 2: while this is definitely an interesting result which carries important messages, it's not immediately clear to me how useful it can be in practice.  The reason is that one wouldn't normally expect \gamma (the expected marginal) to be close to 1.  If, say, \gamma is 1/2 (which is still quite large --- generally one would expect this quantity to decrease as n increases), then the best bound thm 2 can give is p <= 1/2, which wouldn't be particularly helpful.  So my interpretation of thm 2 is: only in extreme cases one can make statistical conclusions based on the expected marginals, and in those cases the importance of the corresponding feature is probably so clear that one wouldn't even need statistical guarantees.  I'm curious what the authors have to say about this.

"Implications of thm 2" paragraph, "\gamma provides both a lower bound and an upper bound": hmm, but (11) is meaningful when \gamma > 0, and (12) is meaningful when \gamma < 0, so the lb and ub are never simultaneously meaningful...

Cor 1 and cor 2: these results are somewhat similar to thm 2 in that one would need a really large Shapley value to say anything meaningful.  (Of course the results do carry conceptual messages.)

Sec 4, synthetic data: here the power of 1 - \gamma is fairly impressive, but it's also clear that a patch with a cross given other patches with noise fully explains the label.  Ideally it would be nice to have a synthetic dataset that appears harder to explain.

Sec 4, real data: I like this part especially because it shows both the power and the limitations of the theory.

**Strengths And Weaknesses:**

- Strengths

The paper is very well written and polished.  The topic is of theoretical and practical importance.  The paper has a clear message, which is Shapley values are connected to conditional independence tests in a formal way.  The results may trigger further research.


- Weaknesses

While the bounds are conceptually meaningful, it's not immediately clear how practical they are.  Also see detailed comments for some minor suggestions.

---

> ### Author Response · Authors · 2023-10-18
> **Response (Part 1)**
>
> * **Medicine example in introduction: Clarify meaning of example**
>
> We thank the reviewer for this question. The intention of the example is to show that currently, there is not a principled way to perform statistically-valid feature selection by means of the Shapley value (i.e., that guarantees Type I error control). As the reviewer suggested, one could decide to report all features with a positive Shapley value as important. However, these is no way of knowing whether this strategy is going to overreport features, which could be misleading with respect to the decision process of the predictor. In the same way, the conservative strategy of reporting only the most important feature will likely underreport findings. We have made this clearer in the revised version of the manuscript
>
> ---
>
> * **First sentence in Sec. 2.1: Reference to cooperative game theory**
>
> We have changed the format and the reference to *"Introduction to the Theory of Cooperative Games"* by Peleg and Sudholter in the revised version of the manuscript.
>
> ---
>
> * **The Shapley value of a feature (i.e., $\phi_j$) can be negative according to Eq. (3)**
>
> We thank the reviewer for highlighting this difference. We have clarified it in the revised manuscript. We would also like to comment that in machine learning explainability, it is common to think of features with a negative contribution as *hurting* the prediction of a model. In fact, Corollary 2 shows that a negative Shapley value implies a *lower*-bound to the corresponding global *p*-value.
>
> ---
>
> * **There as to be some constraint on $C^\*$ in the sentence after Eq. (3)**
>
> We agree with the reviewer. We have clarified this in the revised version of the manuscript. We note the choice of what subset one might be interested in reporting is not unique and may change across domains and data distributions.
>
> ---
>
> * **Clearly state connection of SHAP-XRT with CRT in Sec. 3.1**
>
> We thank the reviewer for this comment. We have included a discussion of the connection of SHAP-XRT with the CRT in the first paragraph of Sec. 3.
>
> ---
>
> * **General criticism of Theorem 2: Practical utility**
>
> We thank the reviewer for raising this important consideration. Theorem 2 does imply the range for which one can make statistically valid claims by means of marginal contributions is indeed narrow. However, we do not consider this a limitation of the theory, but rather of the way these hypotheses are tested within the Shapley value. We would like to remark a few points:
>
> 1. Without this result, it would not be possible to make claims about statistical importance even in the presence of clear effects (i.e., $\gamma_{j,C} \approx 1$).
>
> 2. Theorem 2 can be deployed via Corollary 1 to construct the single statistically valid feature in cases where one does not know it. Suppose there are two features $j,k$ with $\phi_j \approx \phi_k \approx 1/2$. It is true that neither of them is statistically valid on its own, but their union will probably be. This result is of practical relevance, as shown in the real image data experiment, where one can detect sick cells by means of the Shapley value.
>
> 3. We hypothesize the narrow range of Theorem 2 may come from the fact that the marginal contributions in the Shapley value only consider differences in first moment (i.e., means).
>
> 4. We see this results as a stepping stone for future efforts to: $(i)$ devise more powerful ways of combining marginal contributions in a statistically-valid way, and $(ii)$ to reframe other explanation methods in terms of conditional independence testing that may consider higher moments of the distribution of the response of the model (i.e., beyond marginals and means).
>
> ---
>
> * **Implications of Theorem 2: Upper- and lower-bounds do not hold simultaneously**
>
> This is a great point! The upper bound in Eq. (11) still holds in Eq. (12), but a negative $\gamma_{j,C}$ would lead to a vacuous upper bound (i.e., grater than 1). We have removed *"both"* from the sentence to reflect this.
>
> ---
>
> * **Corollary 1 and 2: Only large Shapley values are meaningful**
>
> Yes, this is correct, and related to the comments regarding the limitations of Theorem 2 raised above.

---

> ### Author Response · Authors · 2023-10-18
> **Response (Part 2)**
>
> * **Synthetic image data: Example that is harder to explain**
>
> We thank the reviewer for their comment. The synthetic image data experiment was designed to present a case where SHAP-XRT can work well without having access to the ground-truth response. We included the real image data experiment to show the limitations of testing via the Shapley value in practical scenarios.
>
> As suggested by the reviewer, we included a new supplementary figure to the known response function experiment to better showcase the gap between the SHAP-XRT $p$-value and rejecting by means of the marginal contributions. In particular, we estimate $\mathbb{P}[\gamma_{j,C} \geq 1 - \alpha \mid \hat{p}^{\text{S-XRT}}_{j,C} \leq \alpha]$ for 100 equally-spaced values of $\alpha$ in $[0.05, 1]$ over 100 draws of the SHAP-XRT $p$-values for a fixed dataset of 5,000 samples from the known data distribution. That is, the probability of rejecting by means of marginal contributions at level $\alpha$ when we would have rejected by means of the SHAP-XRT $p$_value. We found that using marginal contributions underreports findings for practical values of $\alpha$ by order of magnitude.
>
> ---
>
> * **Real data experiment**
>
> We thank the reviewer for their feedback! This experiment was included to showcase how one could deploy these ideas in real-world scenarios, and its current limitations.

---

> > ### Comment · Reviewer_1xWV · 2023-10-31
> >
> > Thanks for the response and the revision, which I find satisfactory.  I don't have further questions at the moment.

---

> > > ### Author Response · Authors · 2023-11-06
> > > **Thank you for your comments!**
> > >
> > > We sincerely thank the reviewer for their comments, and we are glad to hear the reviewer finds our response satisfactory.

---

### Review · Reviewer_Jf7B · 2023-08-02

**Summary Of Contributions:**

This work aims to understand Shapley values through a new perspective: their connection with existing, formal hypothesis tests for conditional feature independence. The view shared here is that Shapley values are related to a new, per-sample hypothesis test of conditional feature independence, which the authors call the "Shapley Explanation Randomization Test" (SHAP-XRT). In addition to being "local" (focused on a single prediction), the test is also based on a model rather than the underlying data distribution, and it tests for independence when conditioned on an arbitrary subset of features (vs the more common choice of all the remaining features).

This test is not used today, but the authors present it in some generality and make provable claims about the test's validity. Regarding the specific connection with Shapley values, they claim that the Shapley values in some cases provide upper and lower bounds on the test's p-value. They also consider how to reason jointly about the exponential number of tests that can be performed for each feature (a "global" hypothesis test, not meant in the conventional interpretability sense).

Finally, the authors present a brief set of experiments demonstrating their theory.

**Audience:**

Yes

**Broader Impact Concerns:**

No concerns about broader impacts.

**Claims And Evidence:**

Yes

**Requested Changes:**

Some assorted points about related work:
- In the introduction, perhaps "interpretable models" should be "inherently interpretable models" for clarity?
- In the introduction, the authors write that the Shapley value properties may not be satisfied "perturbation-based" explanations. Many people think of SVs as a perturbation-based method, so I'm not sure I get this. Its properties aren't satisfied by any other method (this follows trivially from the axiomatic characterization), whether perturbation- or gradient-based.
- In the introduction, when alluding to the various approximation strategies, the implication that many methods aren't provable is a bit surprising. For example, KernelSHAP, permutation sampling, TreeSHAP and LinearSHAP are all provably correct (in some cases asymptotically). Also, this is a somewhat sparse list of approximations, a more comprehensive discussion is provided in a recent review paper [1]. (This point also comes up later in section 2.2.)
- In the related work, Data Shapley and its distributional version aren't exactly relevant here, as they're about data sample contributions rather than feature contributions.
- In the related work, there are other papers that discuss information theoretic interpretations of the Shapley value. For example, SAGE [2], which is built on by Watson et al (2023), is based on an information-theoretic view that's more relevant to conditional independence testing. L-/C-Shapley [3] is another example, but that perspective is arguably more about efficient approximation than interpretation of the values.
- In the related work, it could be worth mentioning the other explanation methods based on each feature's marginal contributions.

[1] Chen et al, "Algorithms to estimate Shapley value feature attributions" (2023)

[2] Covert et al, "Understanding global feature contributions with additive importance measures" (2020)

[3] Chen et al, "L-Shapley and C-Shapley: Efficient model interpretation for structured data" (2019)


Assorted presentation points:
- In eq. 5, this looks like a valid way to rewrite eq. 4, but I wonder if it makes more sense to write via $Y$'s conditional distributions? I.e., $(Y \mid X_j, X_{-j}) = (Y \mid X_j)$? This seems more aligned with how Shapley values work, because $f$ approximates $p(Y \mid X)$.
- I personally found SHAP-XRT difficult to follow at first. I'm not sure how the clarity can be improved, but perhaps it could help to emphasize at the beginning of section 3.1 that the test is distinct from and not trivially related to the Shapley value.
- One of the points of confusion about SHAP-XRT, which is perhaps a strength, is that it's formulated as a test of equality in distribution. This was confusing because the standard Shapley value is based on differences in conditional expectations. I wonder if the authors could point out this distinction earlier and more explicitly.
- About Lemma 1 and the ensuing discussion: it seems like we could calculate the global p-value with arbitrary weighting schemes and find whether we reject it under any circumstance. This seems equivalent to testing each of the individual hypotheses, so it surely would require some form of correction. Is the idea then that you must fix your weighting in advance to avoid multiple hypothesis testing that requires corrections?



There were a couple ways in which I wondered how the authors' perspective could be broadened beyond the standard Shapley value:

1. SVs are one of many methods based on a feature's marginal contributions to a coalitional game, see [2]; besides the Banzhaf value, this includes any "semivalue" (studied by Kwon & Zou, 2022), as well as any "random-order value" (studied by the asymmetric Shapley values paper). This could be mentioned with a small modification to the section on global testing.

2. The current analysis is focused on a model $f$ rather than the underlying data distribution. However, it might not be unreasonable to assume that $f$ is close to the true conditional probability. In that case, could the SHAP-XRT be interpreted as a test of true conditional independence vs conditional independence via a model? This perspective is explored in most prior works that discuss information-theoretic interpretations, for example SAGE and [4].

3. The analysis here focuses on Shapley values applied in a "local" setting, hence the need for a local conditional independence test (SHAP-XRT). But the Shapley value is also sometimes used "globally," or across the entire dataset. For example, people heuristically calculate the mean absolute Shapley value, and SAGE provides a more principled global Shapley value via the model's loss. Could either of these correspond to a dataset-level version of the SHAP-XRT?

[4] Covert et al, "Explaining by removing: a unified framework for model explanation" (2021)


About the experiments: I paid more attention to the previous sections, but one detail that jumped out was the choice to use H-SHAP from Teneggi et al (2022) for the real data experiments. I'm not that familiar with the method, but based on my understanding it's not clear that the method actually calculates Shapley values: I thought it returned a subset of relevant features and therefore yielded only coarse approximations, so that it would be biased in most cases. With that in mind, why not use more established options like KernelSHAP or permutation sampling (Mitchell et al., 2022), which are consistent statistical estimators? It would be helpful to hear the authors' explanation for this.

**Strengths And Weaknesses:**

As someone who reads about and uses Shapley values, I'm interested in this line of work and have suspected that such a connection with conditional independence testing exists. I'm glad the authors tackled this problem, and I'm mostly satisfied with their execution (some requests for changes are made below).

Some assorted strengths:
- The motivation for connecting Shapley values with conditional independence testing is well articulated.
- The overview/introduction to Shapley values was easy to read and mostly comprehensive.
- The SHAP-XRT hypothesis test is a non-trivial view to come to about what the Shapley value is "really" doing. The authors presented it in generality, where the conventional Shapley procedure emerges as a special case, and the connection with conditional independence testing via equality in distribution (vs equality in expectation, see eq. 6) was surprising to me.
- The authors gave proper consideration to the test's validity. (I'm not an expert on this topic so I did not verify their proof.)
- The results showing how the Shapley value bounds the tests' p-values, at least in the binary classification setting, was interesting and surprising to me.
- The authors made an attempt to generalize their perspective beyond the Shapley value by mentioning other "probabilistic values" (e.g., Banzhaf). (As mentioned below, I wonder if the authors could have considered other extensions to their perspective.)

---

> ### Author Response · Authors · 2023-10-18
> **Response (Part 1)**
>
> * **"Interpretable models" vs "Inherently interpretable models"**
>
> In saying *"interpretable models"* we were indeed talking about the models that are inherently interpretable. We have made the appropriate changes in the text.
>
> ---
>
> * **SHAP is considered a perturbation-based explanation method**
>
> We thank the reviewer for pointing this out. In the specific sentence, we were referring to gradient-based explanation methods. We have rephrased the sentence accordingly to stress that the Shapley value is the only explanation method that satisfies these theoretical properties.
>
> ---
>
> * **Claim that "many [Shapley value approximation] methods are not provable"**
>
> The reviewer raises a fair point. The message we were trying to convey was that although most approximation strategies are consistent or provide some optimality guarantee, only a few provide finite-sample, non-asymptotic, and efficiency guarantees. We have rephrased the sentence in the introduction, included the review paper by Chen et al., and removed the sentence from Sec. 2.2 for the sake of conciseness.
>
> ---
>
> * **Data/Distributional Shapley as related works**
>
> It is true that Distributional Shapley is defined for data attribution rather than feature attribution. However, we consider the idea of including distributional information relevant within the context of granting the Shapley value with statistical meaning. We have rephrased the sentence in the manuscript to clarify this distinction.
>
> ---
>
> * **Related works that discuss information-theoretic perspectives of the Shapley value**
>
> The reviewer is correct that there exist other works that study information theoretic interpretations of the Shapley value. We would like to remark a few distinctions about the presented work:
>
> 1. SAGE studies global feature importance (i.e., over a population) rather than local feature importance (i.e., on a single sample).
> 2. For local feature importance, Watson et al. study the conditional independence between the true response $Y$ and the features. In this work, we study the conditional independence between a fixed, deterministic model $f$ and the features. That is, the results of Watson et al. consider the case where $f$ is the true conditional probability (i.e., the Bayes' optimal predictor for binary classification), whereas we do not make any assumptions on how well $f$ approximates the true response.
>
> We have included these comments in the related work section of the revised manuscript.
>
> ---
>
> * **Include other explanation methods based on each feature's marginal contributions in related works**
>
> We have complemented with other explanation methods based of features' marginal contributions in the introduction.
>
> ---
>
> * **Different ways to rewrite the CRT null hypothesis in Eq. (4)**
>
> The reviewer is correct that $(Y \mid X_j, X_{-j}) \overset{d}{=} (Y \mid X_{-j})$ is equivalent to Eq. (4). We decided to adopt the notation that is most common in seminal works on conditional independence (Candes et al. [2018], Tansey et al. [2022]). We have included a footnote to clarify this.
>
> Regarding the point that *"$f$ approximates $p(Y \mid X)$"*, we would like to remark that our results are more general in the sense that they do not rely on how good of an estimate $f$ is. This is a fundamental difference between the CRT null hypothesis (i.e., Eq. (4)), and the SHAP-XRT null hypothesis (i.e., Eq. (6)). In the latter, $f$ is a fixe deterministic function of the features, and not a placeholder for $Y$ (which is a random variable of the features).
>
> ---
>
> * **Improve clarity of SHAP-XRT presentation in connection to the Shapley value**
>
> We thank the reviewer for their thoughtful consideration. We have clarified this fundamental difference in the introductory paragraph of Sec. 3.
>
> ---
>
> * **Accounting for multiple-hypotheses in the global $p$-value: weighting schemes**
>
> This is a great question! Note we are not trying to reject each individual hypothesis, but instead the global hypothesis. Given $k$ $p$-values: $p_1, \dots, p_k$, from the individual null hypotheses, there are in fact many ways one could combine these pieces of evidence to reject the global null (i.e., the intersection of the $k$ individual nulls). We will briefly mention 2 methods (and refer to Vovk and Wang [2020], for an in-depth analysis of the many more that exist):
>
> 1. The most straightforward method is Bonferroni correction. One calculates the minimal $p$-value and, if it less than $\alpha/k$, rejects the global null.
> 2. Weighted averaging is also a valid method to combine $p$-values (as long as, as noted by the reviewer, the weighting scheme is fixed *a-priori*). That is $p = \sum_{i \in [k]} w_i p_i$ with $\sum_{i \in [k]} w_i = 1, w_i \geq 0$. This is precisely what the Shapley values do, as shown in Lemma 1.
>
> In general, one may use different weighting schemes that reflect *a-priori* knowledge about the importance of the individual $p$-values towards the global null.

---

> > ### Author Response · Authors · 2023-10-18
> > **Response (Part 2)**
> >
> > * **Semivalues and random order values**
> >
> > We have expanded on the semivalues and random order values in the global hypothesis testing section to include the suggested references.
> >
> > ---
> >
> > * **Could SHAP-XRT be interpreted as a test of true conditional independence?**
> >
> > The reviewer is correct that our analysis is focused on a model rather than the underlying data distribution. The reason is because we are interested in interpreting and understanding a particular model's output for a specific sample $x$. If we assume that $f$ approximates the conditional probability $p(Y = 1 \mid X)$, then yes, SHAP-XRT can be interpreted as a test of true conditional independence. For a fixed $j \in [n]$ and subset $C \subseteq [n] \setminus \\{j\\}$, the null hypothesis would be $H^{\text{S-XRT}}_{0,j,C}:~p(Y = 1 \mid x_C, x_j, \tilde{X} _{-\\{C \cup \\{j\\}\\}} ) = p(Y = 1 \mid x_C, \tilde{X} _{-C})$. However, we do want to note that even if we make this assumption, our method and perspective is different from SAGE because SAGE considers global importance.
> >
> > ---
> >
> > * **Could global notions of importance correspond to a dataset-level version of the SHAP-XRT?**
> >
> > Yes! There exist tests such as the conditional randomization test (CRT) and the holdout randomization test (HRT) which are dataset/population tests of conditional independence. These are related to but different from the SHAP-XRT, in particular with respect to the subsets considered. We believe there could be principled ways to deploy global feature attribution methods to test for conditional independence.
> >
> > ---
> >
> > * **Why not use more established options in the real-world data experiment?**
> >
> > We thank the reviewer for their question! The purpose of Fig. 4 is tho showcase how the theoretical contribution of Theorem 2 translates to a real-world application. In particular, the objective is to compare two cases where the same object of interest (i.e., a sick cell) receives two different Shapley values (i.e., 1 in Fig. 4a, and 0.5 in Fig. 4b) and how this affects the $p$-values of the individual SHAP-XRT null hypotheses. To achieve this, we need to compute each marginal contribution exactly. We stress that the exact computation of each marginal contribution is fundamental for the comparison to hold. We use h-Shap because it precisely satisfies our desiderata. In particular, it considers quadrants instead of pixels to make exact computation tractable. We note that:
> >
> > 1. In this setting, these is no need for permutation sampling because there are only $2^4 = 16$ subsets to evaluate, and
> > 2. KernelSHAP directly approximates Shapley values (i.e., $\phi_j$), whereas we meed access to the individual marginal contributions (i.e., $\gamma_{j,C}$, for each $C \subseteq [n] \setminus \\{j\\}$).
> >
> > We have made this clearer in the revised version of the manuscript.

---

> > > ### Comment · Reviewer_Jf7B · 2023-10-30
> > > **Thanks for revisions**
> > >
> > > The revisions described here all seem helpful, thanks to the authors for responding to the reviewers' feedback. This has improved my rating of the paper, but nonetheless a couple nits the authors might consider:
> > >
> > > - The reference to Data Shapley in Section 1.1 still seems like an odd choice. The feature attribution Shapley value and data valuation Shapley value are very different, and the ways to incorporate distributional information are very different as well. In the feature attribution context, it might make more sense to cite Aas et al 2021 [1] or Frye et al 2021 [2], particularly because it's the use of the conditional distribution that enables connections to information theory.
> > > - Thanks for the explanation re: H-SHAP, I now understand why it was used in the experiment. However, using quadrants leads to a very small number of "features," so I wonder if it would be possible to further subdivide the image, perhaps into 16 grid-shaped superpixels? This leads to only $2^{16} \approx 65000$ subsets, which seems tractable and could lead to more visually meaningful results.
> > >
> > > [1] Aas et al, "Explaining individual predictions when features are dependent: More accurate approximations to Shapley values" (2021)
> > >
> > > [2] Frye et al, "Shapley explainability on the data manifold" (2021)

---

> > > > ### Author Response · Authors · 2023-11-06
> > > > **Thank you for your response!**
> > > >
> > > > We thank the reviewer for their consideration of our revised manuscript. We have now:
> > > >
> > > > * Included the two suggested references in Sec. 2.2 regarding the need to mask features with their conditional distribution in order to keep samples within-distribution, and
> > > >
> > > > * Extended the real image data experiment to include a case with 16 features in Appendix E

---

### Review · Reviewer_u8L8 · 2023-10-28

**Summary Of Contributions:**

In this research, the authors explore the relationship between the Shapley value and conditional hypothesis testing to interpret machine learning models at the sample-specific level. Through their work, they demonstrate that these two methods are intricately connected, with the Shapley value involving the computation of specific conditional hypothesis tests, and each component of the Shapley value providing a means to bound the p-values of these tests. This novel perspective assigns a precise statistical interpretation to the Shapley value, shedding new light on its meaning. The authors conducted numerical experiments using synthetic and real data, offering practical evidence to complement their theoretical results.

**Audience:**

Yes

**Claims And Evidence:**

Yes

**Requested Changes:**

It is better to put footnote 6 into the main content to replace Eq. 5 since it is easier to understand.

**Strengths And Weaknesses:**

**Strengths**
- This paper is well-written and well-structured, the methods are explained very clearly, and the experiments also make sense.
- The idea of connecting Shapley Value and CIT to create an S-CIT method is very intriguing. While previous methods have explored this aspect, the approach in this paper relaxes the assumptions of previous methods and yields solid and fascinating results.
- The experiments in this paper are also well done, including simulations, synthetic data, and real data. The experimental results strongly support the methods proposed in the paper.
- I believe this paper can be accepted with minor revisions. I didn't find any issues with the methods in this paper, and I found the motivation and method of this paper to be very clear.

**Weaknesses**
- As for the contribution, I am only familiar with this topic and not very deeply. Therefore, please refer to the opinions of other reviewers regarding the contribution section.
- I have a concern that the author mentioned in the introduction that the interpretability of deep neural network features is very poor. My understanding of features is that when dealing with high-dimensional data, we first need to use neural networks to extract features to obtain a representation Z=Embedding(X). I believe the poor interpretability arises from our lack of knowledge about the information contained in Z. However, I noticed that in this paper, the concept of features is directly in the observed space of X, which is different from my understanding.

---

> ### Author Response · Authors · 2023-10-30
> **Response**
>
> We thank the reviewer for their time and consideration of our submission!
>
> * **Semantics of feature**
>
> We agree with the reviewer that neural networks are used to extract meaningful representations of high-dimensional data. In fact, following the notation presented in the submission, the classifier $f(x)$ can be seen as the composition $g(h(x))$, where $z = h(x)$ is the embedding of the input $x$, and $g(z)$ is a binary classifier. All results presented in the submission can be directly applied to test for the conditional independence of $g(z)$. It is common to consider feature importance in the space of $x$ because it allows to produce saliency maps that are intelligible to humans. Alternative approaches exist that, for example, explain the predictions of a model in a different domain (e.g., in the wavelet basis [1]).
>
> ---
>
> * **Footnote 6**
>
> We have swapped the two equations in the revised version of the manuscript.
>
> ---
>
> We would be happy to address any outstanding questions.
>
> ---
>
> References:
>
> [1] Stefan Kolek, Duc Anh Nguyen, Ron Levie, Joan Bruna, and Gitta Kutyniok. Cartoon explanations of image classifiers. arXiv preprint arXiv:2110.03485, 2021.

---

> > ### Comment · Reviewer_u8L8 · 2023-10-31
> > **A further concern**
> >
> > Thanks for the response!
> >
> > I think that extending the method described in this paper to traditional features may not be so easy. First, a well-trained classifier tends to extract features only related to the task of classification, that is, features related to the labels. Therefore, in the feature space, irrelevant features may be discarded already.

---

> > > ### Author Response · Authors · 2023-11-06
> > > **Thank you for your suggestion!**
> > >
> > > We thank the reviewer for their valuable comment and suggestion, which will motivate future work to deploy the results presented in this submission to study conditional independence in the embedding space of a classifier rather than in the input space.

---

### Author Response · Authors · 2023-10-18
**Thank you for your feedback!**

We sincerely thank all reviewers for their time and careful consideration of our manuscript. We feel the reviewers' comments and suggestions helped us improve the presentation of our work and of its practical implications.

We have addressed all questions and suggestions raised by the reviewers in our response and in the revised version of the manuscript. We have included a highlighted version of the revised manuscript in the supplementary material, alongside the code that was used to run all experiments presented in the paper.

We are looking forward to engaging with the reviewers to discuss our response and would be more than happy to address any outstanding questions they might have!

---

### Decision · Action_Editor_ANSo · 2023-11-22

**Recommendation:** Accept as is

**Comment:**

This paper explores the intersection of Shapley-based explanation methods and conditional independence testing, traditionally considered separate in the context of artificial neural networks (ANNs). The authors introduce the SHAPley-EXplanation Randomization Test (SHAP-XRT), demonstrating its relevance for local conditional independence. They establish that, for binary classification problems, Shapley values provide bounds for p-values in conditional independence tests. The work advances our understanding of Shapley-based methods, revealing conditions for statistically valid claims about feature importance using the Shapley value.

 The reviewers raised several concerns on the the technical details and presentation, and these concerns were well addressed in the rebuttal. The updated manuscript was recommended for acceptance by all reviewers.

**Audience:**

Yes

**Claims And Evidence:**

Yes